# THE UNREASONABLE EFFECTIVENESS OF PRETRAINING IN GRAPH OOD

## ABSTRACT

Graph neural networks have shown significant progress in various tasks, yet their ability to generalize in out-of-distribution (OOD) scenarios remains an open question. In this study, we conduct a comprehensive benchmarking of the efficacy of graph pre-trained models in the context of OOD challenges, named as `PODGenGraph`. We conduct extensive experiments across diverse datasets, spanning general and molecular graph domains and encompassing different graph sizes. Our benchmark is framed around distinct distribution shifts, including both concept and covariate shifts, whilst also varying the degree of shift. Our findings are striking: even basic pre-trained models exhibit performance that is not only comparable to, but often surpasses, specifically designed to handle distribution shift. We further investigate the results, examining the influence of the key factors (e.g., sample size, learning rates, in-distribution performance etc) of pre-trained models for OOD generalization. In general, our work shows that pre-training could be a flexible and simple approach to OOD generalization in graph learning. Leveraging pre-trained models together for graph OOD generalization in real-world applications stands as a promising avenue for future research.

## 1 INTRODUCTION

Graph Neural Networks (GNNs) have emerged as a popular innovation, primarily due to their unparalleled proficiency in processing graph-structured data (Kipf & Welling, 2017; Wu et al., 2020). However, their performance is markedly diminished when dealing with Out-of-Distribution (OOD) tasks in which training and test data follow different distributions (Li et al., 2022a). The OOD challenges in graph learning have prompted the development of myriad solutions (Yu et al., 2023; Wu et al., 2022a; Feng et al., 2020; Li et al., 2022b; Fan et al., 2022). These methodologies, however, often cater to specific OOD scenarios, such as distinctive data shifts or semantics, making them less versatile due to the dynamic nature of real-world applications.

Given the adaptability and generalizability of pre-trained models in other domains such as images (Kim et al., 2022; Naganuma & Hataya, 2023; Yu et al., 2021; Gulrajani & Lopez-Paz, 2020), there is a potential that graph pre-trained methodologies could significantly enhance the performance of GNNs in addressing graph OOD challenges (Xia et al., 2022). Motivated by this potential, we seek to investigate whether graph pre-trained models can serve as robust and efficient solutions for graph OOD generalization.

In this paper, we systematically investigate the importance of pre-trained models for graph OOD generalization. We consider a variety of graph pre-trained models and diverse distribution shifts. Specifically, we evaluate methodologies such as context prediction (Hu* et al., 2020), mask pre-training learning (Hu* et al., 2020; Xia et al., 2023) along with contrastive learning (Sun et al., 2020). We evaluated their efficacy across various graph datasets, including molecular and general simulated graphs, while adjusting the types of distribution shifts (e.g., covariate shift and concept shift), as well as different distribution shift degrees. Our aim is to empirically verify whether these pre-trained models can achieve better performance in comparison to the state-of-the-art methods specifically designed for OOD problems. Additionally, we explore the implications of the key factors of pre-trained models in OOD contexts, such as sample size, fine-tuning learning rate, in-distribution (ID) learning performances. Fig. 1 summarizes the key components of `PODGenGraph` benchmark. Our key findings, based on various evaluation protocols and analysis, include:

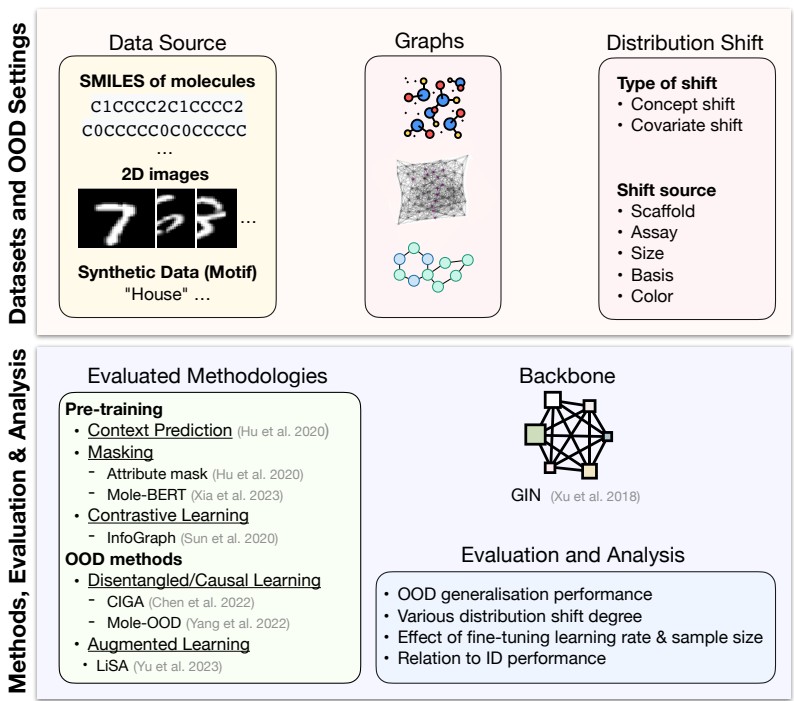

Figure 1: Summary of `PODGenGraph` benchmark.

- In most molecular graph OOD generalization experiments, pre-training methods achieve comparable, mostly superior to specialized OOD methods like invariant/causal learning and graph augmentation (achieving highest or second-highest performances among all 19 datasets). Moreover, pre-trained methods consistently exhibit superiority across a set of degrees of distribution shift, highlighting a simplistic and practical solution for graph OOD, especially in the molecular domain.

- We observe that even with a smaller fine-tuning sample size, such as only 10%-20% of the original fine-tuning sample size, pre-trained models can still achieve comparable results in OOD generalization to those with the full sample size, demonstrating the sample efficiency.

- In general graphs, particularly under concept shift, pre-training methods do not exhibit significant superiority, suggesting that pre-trained models for graph OOD are not universally better, leaving room for future improvements. A possible approach is to combine pre-trained models with other solutions like augmentation and invariant learning to devise more universal algorithms to resolve graph OOD issues.

- Contrary to (Miller et al., 2021), we find that in-distribution learning performance is not always an indicator for OOD generalization, specifically in the context of pre-trained models for graph OOD. This finding might lead to more comprehensive algorithms or theoretical analysis exploring the correlation between OOD and ID learning performance in the future.

- Similarly, different from previous works (Yu et al., 2021; Li et al., 2019), we discover that smaller learning rates during the fine-tuning phase do not invariably lead to better generalization in OOD scenarios for most pre-trained graph models, with the exception being the mask pre-training method Molecule-BERT that introduces prior information.

**Contributions**: 1. We execute an extensive evaluation of various graph pre-training strategies such as masked pre-training, contrastive learning, graph auto-encoders, and context prediction, against multiple types of distribution shifts including covariate and concept shifts.

2. We explore the impacts of different key factors in pre-training, such as sample size, fine-tuning learning rates, and in-distribution learning performances, alongside varying degrees of distribution shift, to disclose their implications on the effectiveness of pre-trained models in OOD generalization.

3. Our key findings of the `PODGenGraph` benchmark could potentially offer valuable insights for future works regarding the selection of pre-trained models and the development of more advanced methodologies for graph OOD.

## 2 RELATED WORKS

**Graph Pre-training.** Graph pre-training with external datasets has been explored in numerous studies but there is no specific study on evaluating them under OOD generalization context. Slightly different from self-supervised methods for OOD generalization, pre-training methods usually leverage the external datasets to learn the good initlization or representation which could benefit the downstreaming tasks. Here we summarize the current state-of-the-art approaches in two categories: supervised and self-supervised pre-training methods. (1). **Supervised Pre-training**. While supervised labels often require significant time and resources, they can still aid pre-training, particularly in biochemical contexts. Hu* et al. (2020) utilized these to predict a plethora of molecular properties and protein functions. They also considered structural similarities between graphs as a form of supervision. MOCL (Sun et al., 2021) further explored this by measuring the structural similarity between molecules via the Tanimoto coefficient. Other methods like GROVER (Rong et al., 2020) and MGSSL (Zhang et al., 2021) were introduced to predict the motifs in molecular graphs. (2).**Self-supervised Pre-training.** Graph AutoEncoders (Kipf & Welling, 2016) aim to reconstruct parts of graphs that aid in understanding the data representation. Graph Autoregressive Models, such as GPT-GNN (Hu et al., 2020) and MGSSL (Zhang et al., 2021) use the autoregressive framework for graph reconstruction. Masked components modeling, including the mask pre-training strategy in (Hu* et al., 2020) and Mole-BERT (Xia et al., 2023), utilizes masking components from graphs and then predicting them. InfoGraph (Sun et al., 2020) and DGI (Veličković et al., 2018) employ mutual information maximization between various graph representations. GraphCL (You et al., 2020) introduces a contrastive learning framework emphasizing robust, transferable representation learning with graph augmentations for enhanced generalizability. In our work, we choose the self-supervised pre-training methods for evaluation due to the practical considerations. We expand upon their OOD generalization results and analysis by conducting a thorough exploration of graph pre-trained models. Our study goes beyond the confines of MoleculeNet datasets with covariate shifts, exploring diverse data sources including DrugOOD (Ji et al., 2023), MoleculeNet (Wu et al., 2018), OGBG (Hu et al., 2020), and others. Furthermore, we investigate distribution shifts with varying degrees of intensity, incorporating various meta-analyses such as the examination of fine-tuning sample size and learning rates to provide a more inclusive analysis. In supervised learning, labels are typically hard to acquire, or the acquisition is highly costly (Wang et al., 2023). In molecular graphs or certain biological graphs, obtaining annotations is challenging. Given our aim is to provide the evlatuion and benchmark for pre-trained models for OOD generalization which could be used for practical real-world problem, we select to evaluate the self-supervised pre-training methods.

**Graph OOD.** The Out-of-Distribution (OOD) generalization problem, well-recognized in machine learning paradigms, gains unique complexities for graph data. Graphs, inherently non-Euclidean, can manifest a broad spectrum of topological structures and dynamics that challenge traditional methods. Three research lines have been conducted to tackle the graph OOD challenge: (1). **Disentangled, Invariant, and Causal Learning.** Disentangled graph representation learning seeks to factorize real-world graphs into distinct latent components. Such models aim to capture underlying, informative factors in the graph data, which has been shown to benefit OOD generalization. The pioneering work of DisenGCN (Ma et al., 2019) introduces a novel convolutional layer, DisenConv, which uses a neighborhood routing mechanism to analyze and infer latent factors. IPGDN (Liu et al., 2020) enhances this by adding an independence regularization to minimize dependencies among representations. FactorGCN (Yang et al., 2020) focuses on graph-level representation, using a factorization mechanism to produce hierarchical disentanglements. Recently, Mole-OOD (Yang et al., 2022), DisC (Fan et al., 2022) and CIGA (Chen et al., 2022b) specifically disentangle causal from non-causal information, offering a robust approach to handle biases and distribution shifts in graphs. These advances spotlight the potential of disentangled representations in achieving superior OOD performance on graph data. (2). **Graph Augmentation.** The structure and topology of graphs play a critical role in predicting their properties. Some methods leverage structure-wise augmentations to generate diverse training topologies. GAug (Zhao et al., 2021) enhances generalization using a differentiable edge predictor, MH-Aug (Park et al., 2021) uses Markov chain Monte Carlo sampling for controlled augmentation. Additionally, feature-wise augmentations

have emerged, where node features are manipulated. GRAND (Feng et al., 2020) randomly drops and propagates node features to reduce sensitivity to specific neighborhoods, while FLAG (Kong et al., 2022) augments node features using gradient-based adversarial perturbations, maintaining the underlying graph structures. LiSA (Yu et al., 2023) further solves the problem of inconsistent predictive relationships among augmented environments by invariant subgraph training. These methods verify the significance of graph data augmentation in achieving enhanced out-of-distribution generalization. (3). **Self-supervised Learning.** Graph self-supervised learning has also shown promise for OOD generalization. For instance, PATTERN (Yehudai et al., 2021) seeks to achieve the generalization from small to large graphs. GraphCL (You et al., 2020) and RGCL (Li et al., 2022d) use contrastive learning, with the latter emphasizing rationale-aware augmentations. Test-time training methods like GAPGC (Chen et al., 2022a) and GT3 (Wang et al., 2022) further innovate by introducing contrastive loss variants and hierarchical self-supervised frameworks, respectively for OOD generalization. Our work is close to this research line, and is the first to discover the universal benefits of self-supervised pre-training to graph OOD, in terms of various graph OOD scenarios. We choose the state-of-the-art methods from the first two research lines for comparison, including CIGA (Chen et al., 2022b), Mole-OOD (Yang et al., 2022), and LiSA (Yu et al., 2023).

## 3 PRELIMINARIES

### 3.1 GRAPH OOD SCENARIOS

We consider both the general feature distribution shifts (*e.g., molecules under different assays*) and structure distribution shifts (*e.g., different molecular size*). Given a training dataset $\mathcal{D}_{\text{train}}$ consisting of $N$ graphs $\{G_1, G_2, ...G_N\}$ each associated with a target label or property $\{y_1, y_2, ...y_N\}$, the graph OOD problem arises when:

$$P(G, y|\mathcal{D}_{\text{test}}) \neq P(G, y|\mathcal{D}_{\text{train}}) \tag{1}$$

In this paper, we consider two types of OOD: covariate shift and concept shift.

**Covariate Shift.** Covariate shift refers to a scenario where the distribution of the input data (graphs in our context) changes between training and test stages, while the conditional distribution of the target given the input remains consistent. Mathematically, if $G$ represents our input graphs and $Y$ represents our labels:

$$P_{\text{train}}(G) \neq P_{\text{test}}(G), \quad P_{\text{train}}(Y|G) = P_{\text{test}}(Y|G) \tag{2}$$

For graph-structured data, covariate shift could imply that while the method of labeling nodes or edges remains consistent, the types of graphs in the test set might differ from those in the training set.

**Concept Shift.** Concept or label shift arises when the distribution of the labels changes between training and testing, even if the input distribution remains the same. Formally:

$$P_{\text{train}}(Y) \neq P_{\text{test}}(Y), \quad P_{\text{train}}(G|Y) = P_{\text{test}}(G|Y) \tag{3}$$

In the context of graph data, this means that while the types of graphs remain consistent across training and test datasets, the manner or criteria by which they are labeled has evolved or changed.

### 3.2 GRAPH PRE-TRAINING METHODOLOGIES

In this section, we briefly discuss the pre-training methods we choose for this study. The detailed training and fine-tuning settings will be discussed in Section 4.2. For molecular datasets, we choose three pre-training methods: ContextPred (Hu* et al., 2020), Attribute masking (Hu* et al., 2020), and Mole-BERT (Xia et al., 2023).

- **ContextPred**: The goal of ContextPred is to pre-train a GNN in such a way that it establishes proximity between embeddings of nodes that occur within analogous structural contexts. It employs subgraphs to predict the surrounding graph structures of these nodes. In this work, we employ the $K$-hop neighborhood as the subgraph in the original work and choose $K = 5$. We also follow the context definition in the work (i.e., adjacent graph structure), and choose the hop values $r_1 = 4$ and $r_2 = 7$.

- **Attribute masking** & **Mole-BERT** & **GraphMAE** : All three works use the masked component modeling methods for the self-supervised learning. Specifically, they involve the masking of certain components within molecules, including atoms, bonds, and fragments, followed by training the model to predict these masked components based on the remaining contextual information. We follow the setups in the original papers: Mask pre-training in (Hu* et al., 2020) inputs atom and chemical bond attributes are randomly masked, and GNNs are pre-trained to predict these masked attributes and Mole-BERT (Xia et al., 2023) uses a context-aware tokenizer that encodes atoms with chemically meaningful values for masking. **GraphMAE** (Hou et al., 2022) represents a significant advancement in the field of graph autoencoders (GAEs). It diverges from traditional GAEs by prioritizing feature reconstruction over graph structure reconstruction and employs a novel masking strategy combined with scaled cosine error, enhancing training robustness and error metric accuracy.

- **GraphCL** (You et al., 2020) introduces an contrastive learning framework focusing on robust and transferable representation learning. It also utilizes the graph augmentations to enhance data priors, to improve the generalizability and robustness.

For general graph datasets as well as the molecular datasets without node information, we use one contrastive self-supervised pre-training, InfoGraph (Sun et al., 2020). It extracts expressive representations for graphs or nodes by maximizing mutual information between graph-level and substructure-level representations at varying granularities.

## 4 BENCHMARK METHODOLOGY

### 4.1 BENCHMARK SETUP

**Datasets.** We evaluate pre-trained models upon multiple dataset sources, including three datasets from DrugOOD (Ji et al., 2023) (`DrugOOD-lbap-core-ic50-assay`, `DrugOOD-lbap-core-ic50-scaffold`, and `DrugOOD-lbap-core-ic50-size`), ten datasets from MoleculeNet (Wu et al., 2018) (BBBP, Tox21, ToxCast, SIDER, ClinTox, MUV, HIV, BACE, OGBG-MolHIV, OGBG-MolPCBA), four datasets from the TU collection (Morris et al., 2020) (NCI1, NCI109, PROTEINS, DD), and three datasets from the general graph collection (Gui et al., 2022) (Motif (Wu et al., 2022b) and CMNIST (Arjovsky et al., 2019)). Table 1 lists the statistics and key factors of the molecular datasets we employed (Detailed version in Appendix Table A1). The detailed statistics of simulated graphs (Motif and CMNIST) is given in Appendix Table A2. We also give the detailed introduction to all datasets in Appendix B.2.

**Various Graph Domains.** We select datasets covering a wide array of graph structures. This includes molecular graphs used in biophysics and physiology research, encompassing both those with and without node information. Additionally, we include synthetic and real-world graphs that represent images and hierarchical trees.

**Source of Distribution Shift and OOD.** We use diverse datasets covering various causes of distribution shift, featuring variations in graph characteristics (like scaffold, size, basis, and color) for both molecular and general graphs, as well as environmental factors (such as assay) for molecular graphs. In the DrugOOD dataset (Ji et al., 2023), the distribution shift originates from disparities in Bemis-Murcko scaffold size (DrugOOD-Scaffold), assay ID (DrugOOD-Assay), and molecular atom size (DrugOOD-Size). In contrast, all datasets within the MoleculeNet (Wu et al., 2018) follow a shift based on the Bemis-Murcko scaffold. For the TU collection (Morris et al., 2020), we utilize data splits generated by (Yehudai et al., 2021) based on molecular atom size. Following (Gui et al., 2022), Motif (Wu et al., 2022b) is tailored to address structural and size shifts, whereas CMNIST (Arjovsky et al., 2019) is partitioned based on different digit colors. We consider both covariate and concept shifts under different domains for most of the datasets.

### 4.2 BASELINES, IMPLEMENTATION, AND EVALUATION

**Baseline Algorithms.** We integrate empirical risk minimization (ERM) (Vapnik, 1999) and the state-of-the-arts with disentangled, invaraint and causal learning, and data-augmentation methodologies. All methods have been reproduced based on their original implementation (details listed in Appendix C.1). We choose two disentangled OOD algorithms, CIGA (Chen et al., 2022b) and MoleOOD (Yang et al., 2022), both based on the invariant and causal learning. CIGA (Chen et al.,

Table 1: **Molecular dataset statistics**. Gray shaded rows indicate datasets without node labels. AP, MCC, and ACC represent the average precision, Matthews correlation coefficient, and accuracy, respectively.

| Datasets | Domain | Shift | #. Graphs | Avg. #. Node | Avg. #. Edge | #. Classes /Task | #. Task | Metrics |
|---|---|---|---|---|---|---|---|---|
| DrugOOD | Scaffold | | 59,608 | 30.0 | 64.9 | 2 | 1 | ROC-AUC |
| | Assay | | 72,239 | 32.3 | 70.2 | 2 | | |
| | Size | | 70,672 | 30.7 | 66.9 | 2 | | |
| BBBP | Scaffold | Covariate | 2,039 | 24.1 | 51.9 | 2 | 1 | |
| Tox21 | | | 7,831 | 18.6 | 38.6 | 2 | 12 | |
| ToxCast | | | 8,575 | 18.8 | 38.5 | 2 | 617 | |
| SIDER | | | 1,427 | 33.6 | 70.7 | 2 | 27 | |
| ClinTox | | | 1,478 | 26.2 | 55.8 | 2 | 2 | |
| MUV | | | 93,087 | 24.2 | 52.6 | 2 | 17 | |
| HIV | | | 41,127 | 25.5 | 54.9 | 2 | 1 | |
| BACE | | | 1,513 | 34.1 | 73.7 | 2 | 1 | |
| OGBG-MolHIV | Scaffold | Covariate | 41,127 | 25.5 | 27.5 | 2 | 1 | |
| | | Concept | | | | | | |
| | Size | Covariate | | | | | | |
| | | Concept | | | | | | |
| OGBG-MolPCBA | Scaffold | Covariate | 437,929 | 26.0 | 28.1 | 2 | 128 | AP |
| | | Concept | | | | | | |
| | Size | Covariate | | | | | | |
| | | Concept | | | | | | |
| NCI1 | Size | Covariate | 2,569 | 27.2 | 58.8 | 2 | 1 | MCC |
| NCI109 | | | 2,500 | 27.2 | 58.8 | 2 | 1 | |
| PROTEINS | | | 679 | 35.8 | 131.2 | 2 | 1 | |
| DD | | | 710 | 244.5 | 1226.8 | 2 | 1 | |
| Motif | Basis | Covariate | 24,000 | 16.6 | 44.7 | 3 | 1 | ACC |
| | | Concept | 24,600 | 17.0 | 48.6 | | | |
| | Size | Covariate | 24,000 | 28.5 | 76.0 | | | |
| | | Concept | 24,600 | 51.7 | 141.4 | | | |
| CMNIST | Color | Covariate | 56,000 | 75.0 | 1393.0 | 10 | 1 | ACC |
| | | Concept | 57,400 | 75.0 | 1393.0 | | | |

2022b) categorizes interactions between causal and non-causal components into fully informative invariant features (FIIF) and partially informative invariant features (PIIF). MoleOOD (Yang et al., 2022) identifies molecule environments without manual specification and uses them along with substructures for predictions. Furthermore, we adopt one augmentation-based OOD algorithm, LiSA (Yu et al., 2023). It utilizes variational subgraph generators to identify locally predictive patterns and generates multiple label-invariant subgraphs, enhancing diversity for data augmentation process. We also consider cases GIN-OOD and GIN-ID, where GIN is trained without specified operations for OOD. GIN-OOD is tested on OOD testing sets, whereas GIN-ID is tested on in-distribution sets.

**Pre-training Datasets.** In accordance with previous works by Hu* et al. (2020), we use 2 million molecules sampled from the ZINC-15 database (Sterling & Irwin, 2015), to learn node representations for downstream molecular datasets. Considering the lack of shared node information across general graph dataset and TU dataset, we initially exclude the label information for self-supervised learning. Once we have learned the representation of each graph, we proceed to fine-tune the classifier (e.g., SVM, logistic regression, or random forest) using a dataset that includes label information.

**GNN Architectures.** We adopt 5-layer graph isomorphism networks (GINs) (Xu et al., 2018) with 300-dimensional hidden units as the backbone model for all pre-training methods in all datasets. The average pooling is used as the `READOUT` function.

**Pre-training and Fine-tuning.** In the pre-training phase, the models undergo 100 training epochs with a batch size of 256 and a learning rate set to 0.001. During the subsequent fine-tuning phase, we conduct training for 100 epochs with a batch size of 32, except for DrugOOD with a batch size of 128, and we report the test score with the best cross-validation performance. In both phases, the models are trained using Stochastic Gradient Descent (SGD) with the Adam optimizer.

### 4.2.1 EVALUATION METRICS

We utilize the original evaluation metrics associated with each dataset. Specifically, in the context of molecular datasets, we report ROC-AUC for DrugOOD and MoleculeNet following Ji et al. (2023); Wu et al. (2018), average precision (AP) for OGBG-MolPCBA following Hu et al. (2020), and the Matthews correlation coefficient for TU datasets following Bevilacqua et al. (2021). Furthermore, for all general graph datasets, we use classification accuracy as our primary evaluation metric.

Table 2: Performance evaluation on different OOD datasets. Different evaluation metrics are employed for different datasets. DrugOOD, MoleculeNem, OGBG-MolHIV: Testing AOC-RUC; OGBG-MolPCBA: Testing Average Precision (AP); TU datasets: Testing Matthews correlation coefficient (MCC); Motif and CMNIST: Testing Accuracy. "cov" and "cpt" denote covariate and concept shift, respectively. Brown shaded columns indicate pre-training strategies. The first and second best-performing numbers (except the ID training) are in **bold** and **bold**, respectively.

| | Methods | GIN-OOD | GIN-ID | CIGA-v1 | CIGA-v2 | MoleOOD | LiSA | ContextPred | AttrMask | Mole-BERT | GraphCL | GraphMAE | InfoGraph |
|---|---|---|---|---|---|---|---|---|---|---|---|---|---|
| DrugOOD | SCAFFOLD (cov) | $67.31_{\pm0.50}$ | $84.36_{\pm0.15}$ | $69.27_{\pm0.81}$ | $69.68_{\pm0.21}$ | $68.01_{\pm0.39}$ | $65.71_{\pm0.25}$ | $70.01_{\pm0.13}$ | $\mathbf{70.68_{\pm0.31}}$ | $\underline{70.04_{\pm0.25}}$ | $68.74_{\pm0.12}$ | $69.37_{\pm0.15}$ | - |
| | ASSAY (cov) | $71.20_{\pm0.29}$ | $87.07_{\pm0.62}$ | $72.36_{\pm0.60}$ | $\mathbf{73.28_{\pm0.35}}$ | $71.18_{\pm0.63}$ | $70.66_{\pm0.63}$ | $\underline{72.80_{\pm0.55}}$ | $71.56_{\pm0.43}$ | $71.19_{\pm0.09}$ | $69.59_{\pm0.10}$ | $70.40_{\pm0.12}$ | - |
| | SIZE (cov) | $66.67_{\pm0.26}$ | $87.69_{\pm0.77}$ | $67.08_{\pm0.82}$ | $68.02_{\pm0.51}$ | $66.61_{\pm0.36}$ | $65.78_{\pm0.46}$ | $\mathbf{68.42_{\pm0.10}}$ | $\underline{68.22_{\pm0.15}}$ | $67.92_{\pm0.19}$ | $67.70_{\pm0.28}$ | $67.97_{\pm0.31}$ | - |
| | AVG. | 68.39 | 86.37 | 69.57 | 70.32 | 68.60 | 67.38 | 70.41 | 70.15 | 69.60 | 68.68 | 69.25 | - |
| MoleculeNet | BBBP (cov) | $65.78_{\pm4.90}$ | $93.13_{\pm0.58}$ | $65.50_{\pm1.62}$ | $68.69_{\pm1.37}$ | $69.71_{\pm1.56}$ | $65.26_{\pm2.01}$ | $69.32_{\pm1.03}$ | $64.95_{\pm3.40}$ | $\mathbf{71.88_{\pm1.12}}$ | $68.02_{\pm1.03}$ | $\underline{71.19_{\pm1.11}}$ | - |
| | Tox21 (cov) | $73.95_{\pm0.28}$ | $82.60_{\pm0.20}$ | $73.87_{\pm0.54}$ | $72.25_{\pm1.46}$ | $73.65_{\pm0.85}$ | $66.32_{\pm0.76}$ | $74.47_{\pm0.36}$ | $\underline{76.22_{\pm0.41}}$ | $\mathbf{77.99_{\pm0.33}}$ | $76.18_{\pm0.50}$ | $76.20_{\pm0.41}$ | - |
| | ToxCast (cov) | $62.13_{\pm0.71}$ | $70.93_{\pm0.28}$ | $62.81_{\pm0.55}$ | $58.53_{\pm1.85}$ | $62.90_{\pm0.96}$ | $59.56_{\pm0.57}$ | $\underline{63.43_{\pm0.40}}$ | $60.15_{\pm0.57}$ | $\mathbf{64.18_{\pm0.31}}$ | $63.31_{\pm0.43}$ | $63.40_{\pm0.41}$ | - |
| | SIDER (cov) | $57.38_{\pm1.65}$ | $62.57_{\pm0.81}$ | $57.40_{\pm4.40}$ | $54.90_{\pm2.13}$ | $\underline{62.01_{\pm0.58}}$ | $57.28_{\pm0.66}$ | $60.45_{\pm0.60}$ | $60.15_{\pm0.57}$ | $\mathbf{62.74_{\pm0.89}}$ | $60.46_{\pm0.98}$ | $60.18_{\pm1.02}$ | - |
| | ClinTox (cov) | $57.29_{\pm5.91}$ | $84.91_{\pm2.10}$ | $55.00_{\pm1.60}$ | $66.37_{\pm3.22}$ | $\mathbf{89.93_{\pm3.90}}$ | $65.00_{\pm2.60}$ | $57.40_{\pm3.16}$ | $70.47_{\pm3.43}$ | $\underline{78.88_{\pm2.24}}$ | $77.53_{\pm3.65}$ | $76.49_{\pm2.95}$ | - |
| | MUV (cov) | $70.40_{\pm1.80}$ | $79.49_{\pm1.44}$ | $68.10_{\pm1.30}$ | $70.99_{\pm1.34}$ | $67.79_{\pm2.46}$ | $67.91_{\pm1.13}$ | $77.36_{\pm1.11}$ | $74.93_{\pm2.07}$ | $\mathbf{78.62_{\pm1.51}}$ | $77.50_{\pm0.61}$ | $\underline{77.51_{\pm1.87}}$ | - |
| | HIV (cov) | $75.06_{\pm2.06}$ | $80.86_{\pm1.11}$ | $75.79_{\pm1.09}$ | $73.19_{\pm4.22}$ | $\mathbf{78.29_{\pm0.51}}$ | $62.57_{\pm1.30}$ | $77.56_{\pm0.95}$ | $76.41_{\pm0.70}$ | $\underline{78.10_{\pm0.65}}$ | $76.81_{\pm0.61}$ | $77.12_{\pm0.54}$ | - |
| | BACE (cov) | $70.78_{\pm5.29}$ | $86.73_{\pm1.72}$ | $73.60_{\pm4.30}$ | $78.56_{\pm2.34}$ | $\mathbf{81.10_{\pm1.97}}$ | $69.97_{\pm3.06}$ | $79.41_{\pm1.96}$ | $\underline{79.88_{\pm0.61}}$ | $80.88_{\pm1.45}$ | $77.96_{\pm2.00}$ | $79.65_{\pm1.40}$ | - |
| | AVG. | 66.70 | 80.55 | 67.75 | 68.16 | 73.36 | 64.92 | 68.38 | 71.37 | 74.62 | 72.22 | 72.72 | - |
| OGBG-PCBA | SIZE (cov) | $12.85_{\pm0.34}$ | $28.10_{\pm0.69}$ | $10.51_{\pm0.17}$ | $9.65_{\pm0.12}$ | - | $6.52_{\pm0.20}$ | $13.30_{\pm0.37}$ | $13.50_{\pm0.38}$ | $\mathbf{16.19_{\pm0.24}}$ | $13.55_{\pm0.31}$ | $\underline{14.17_{\pm0.32}}$ | - |
| | SIZE (cpt) | $12.76_{\pm0.62}$ | $28.10_{\pm0.69}$ | $9.22_{\pm0.09}$ | $8.31_{\pm0.12}$ | - | $5.05_{\pm0.32}$ | $11.39_{\pm0.21}$ | $11.87_{\pm0.24}$ | $\mathbf{15.71_{\pm0.26}}$ | $\underline{12.94_{\pm0.27}}$ | $11.82_{\pm0.17}$ | - |
| | SCAFFOLD (cov) | $13.03_{\pm0.43}$ | $30.80_{\pm0.54}$ | $10.24_{\pm1.98}$ | $10.62_{\pm1.04}$ | - | $8.67_{\pm0.24}$ | $\mathbf{22.14_{\pm0.43}}$ | $\underline{21.89_{\pm0.27}}$ | $17.33_{\pm0.12}$ | $14.91_{\pm0.13}$ | $15.14_{\pm0.15}$ | - |
| | SCAFFOLD (cpt) | $17.27_{\pm0.63}$ | $30.80_{\pm0.54}$ | $8.33_{\pm0.06}$ | $8.71_{\pm0.12}$ | - | $8.55_{\pm0.63}$ | $15.71_{\pm0.38}$ | $16.14_{\pm0.49}$ | $\mathbf{21.29_{\pm0.53}}$ | $\underline{18.85_{\pm0.14}}$ | $17.35_{\pm0.11}$ | - |
| | AVG. | 13.98 | 29.45 | 9.58 | 9.32 | - | 7.20 | 15.63 | 15.85 | 17.63 | 15.06 | 14.62 | - |
| OGBG-HIV | SIZE (cov) | $60.06_{\pm1.63}$ | $79.49_{\pm0.55}$ | $61.81_{\pm1.68}$ | $59.55_{\pm2.56}$ | - | $59.65_{\pm1.44}$ | $60.47_{\pm0.98}$ | $62.29_{\pm0.91}$ | $\mathbf{66.95_{\pm0.93}}$ | $65.86_{\pm1.00}$ | $\underline{66.03_{\pm0.21}}$ | - |
| | SIZE (cpt) | $70.20_{\pm1.12}$ | $79.49_{\pm0.55}$ | $72.80_{\pm1.35}$ | $\underline{73.62_{\pm1.33}}$ | - | $72.36_{\pm4.75}$ | $70.41_{\pm0.38}$ | $70.59_{\pm0.58}$ | $\mathbf{75.94_{\pm0.91}}$ | $72.64_{\pm0.27}$ | $70.85_{\pm0.17}$ | - |
| | SCAFFOLD (cov) | $65.41_{\pm1.70}$ | $80.86_{\pm1.11}$ | $69.40_{\pm2.39}$ | $69.40_{\pm1.97}$ | - | $68.92_{\pm0.92}$ | $70.69_{\pm1.12}$ | $70.29_{\pm1.57}$ | $\mathbf{71.78_{\pm0.96}}$ | $\underline{71.12_{\pm1.21}}$ | $70.61_{\pm1.09}$ | - |
| | SCAFFOLD (cpt) | $62.36_{\pm2.20}$ | $80.86_{\pm1.11}$ | $70.79_{\pm1.55}$ | $71.65_{\pm1.33}$ | - | $69.46_{\pm0.83}$ | $68.77_{\pm0.90}$ | $71.50_{\pm0.55}$ | $\mathbf{76.13_{\pm0.39}}$ | $\underline{73.64_{\pm0.34}}$ | $72.57_{\pm0.77}$ | - |
| | AVG. | 64.51 | 80.18 | 68.70 | 68.56 | - | 67.60 | 67.59 | 68.67 | 72.70 | 70.82 | 70.02 | - |
| TU | NCI1 (cov) | $0.21_{\pm0.06}$ | $0.45_{\pm0.03}$ | $0.22_{\pm0.07}$ | $\underline{0.27_{\pm0.07}}$ | - | $0.24_{\pm0.01}$ | - | - | - | - | - | $\mathbf{0.39_{\pm0.01}}$ |
| | NCI109 (cov) | $0.16_{\pm0.05}$ | $0.44_{\pm0.02}$ | $0.23_{\pm0.09}$ | $0.22_{\pm0.05}$ | - | $0.26_{\pm0.02}$ | - | - | - | - | - | $\mathbf{0.38_{\pm0.01}}$ |
| | PROTEINS (cov) | $0.23_{\pm0.05}$ | $0.46_{\pm0.03}$ | $0.40_{\pm0.06}$ | $0.31_{\pm0.12}$ | - | $\underline{0.43_{\pm0.05}}$ | - | - | - | - | - | $\mathbf{0.53_{\pm0.07}}$ |
| | DD (cov) | $0.25_{\pm0.09}$ | $0.40_{\pm0.04}$ | $0.29_{\pm0.08}$ | $0.26_{\pm0.08}$ | - | $\mathbf{0.37_{\pm0.07}}$ | - | - | - | - | - | $\underline{0.35_{\pm0.04}}$ |
| | AVG. | 0.21 | 0.44 | 0.29 | 0.27 | - | $\underline{0.33}$ | - | - | - | - | - | **0.41** |
| Motif | BASIS (cov) | $62.01_{\pm3.92}$ | $92.15_{\pm0.04}$ | $66.43_{\pm11.31}$ | $67.15_{\pm8.19}$ | - | $\underline{82.55_{\pm7.18}}$ | - | - | - | - | - | $\mathbf{86.85_{\pm2.43}}$ |
| | BASIS (cpt) | $72.12_{\pm1.89}$ | $92.15_{\pm0.04}$ | $72.50_{\pm4.02}$ | $77.48_{\pm2.54}$ | - | $\mathbf{87.89_{\pm1.61}}$ | - | - | - | - | - | $\underline{79.36_{\pm1.12}}$ |
| | SIZE (cov) | $52.94_{\pm2.93}$ | $92.16_{\pm0.07}$ | $49.14_{\pm8.34}$ | $\underline{54.42_{\pm3.11}}$ | - | $\mathbf{62.90_{\pm8.30}}$ | - | - | - | - | - | $53.43_{\pm8.09}$ |
| | SIZE (cpt) | $58.23_{\pm1.73}$ | $92.16_{\pm0.07}$ | $58.63_{\pm6.66}$ | $\mathbf{70.65_{\pm4.81}}$ | - | $\underline{70.36_{\pm2.61}}$ | - | - | - | - | - | $64.79_{\pm1.68}$ |
| | AVG. | 61.33 | 92.16 | 61.68 | 67.43 | - | 75.93 | - | - | - | - | - | $\underline{71.11}$ |
| CMNIST | COLOR (cov) | $26.28_{\pm5.95}$ | $77.80_{\pm0.20}$ | $\underline{32.22_{\pm2.67}}$ | $32.11_{\pm2.53}$ | - | $\mathbf{33.21_{\pm13.43}}$ | - | - | - | - | - | $24.39_{\pm2.09}$ |
| | COLOR (cpt) | $29.53_{\pm0.50}$ | $77.80_{\pm0.20}$ | $34.80_{\pm3.33}$ | $\mathbf{39.39_{\pm3.30}}$ | - | $\underline{36.56_{\pm0.40}}$ | - | - | - | - | - | $19.19_{\pm2.17}$ |
| | AVG. | 27.91 | 77.80 | 33.51 | **35.75** | - | $\underline{34.89}$ | - | - | - | - | - | 21.79 |

We employ 10 random seeds for all methods to get the mean and standard deviation (std) results for each studied baseline. To better evaluate the performance gap among methods, we also consider additional statistical metrics including median and interquartile mean (IQM). Additionally, we also calculate the optimality gap, quantified by the the performance gap between each method and the in-distribution learning one, which ideally serves as the empirical upper-bound result for each task.

## 4.3 RESULTS ANALYSIS

**General Results.** Table 2 gives the results on all evaluated datasets and OOD scenarios. Additionally, Fig. A2-A3 gives the further statistical metrics including median, IQM, mean, and optimality gap across datasets in Drug-OOD and MoleculeNet, respectively. The extensive results reveal that pre-trained methods predominantly outperform methods explicitly designed for Graph OOD tasks across a majority of datasets. Specifically, within molecule-related graph datasets, pre-trained methods achieve the highest or second-highest values all of the 19 test sets, demonstrating the substantial advantages of these methods in such contexts. Among all pre-trained strategies, Mole-BERT consistently performs the best or the second best on most of molecular datasets. This is because that Mole-BERT utilizes a context-aware tokenizer for encoding atoms, which might be more effective in capturing the nuanced chemical properties essential for molecular datasets compared with ContextPred that focuses on predicting the surrounding graph structures of nodes within similar contexts.

In terms of general simulated graphs, pre-trained model-InfoGraph demonstrates performance on the Motif dataset that is comparable to other methodologies (second-highest in average), further underscoring the potential efficacy of pre-trained models in addressing OOD challenges effectively. However, results on CMNIST datasets show that InfoGraph underperforms compared to baseline models, even inferior to GIN-OOD in both concept and covariate shifts. This suggests the pre-trained model's effectiveness on this case isn't superior to other OOD methods, possibly due to the semantic simplicity of the graphs impacting pre-trained representations initializations.

### 4.4 IMPORTANCE OF KEY FACTORS IN PRE-TRAINED MODELS FOR GENERALIZATION

Figure 2: **Analysis on Key Factors in Pre-training.** (a). **Effects of Shift Degree**. Generalization capabilities of all considered methods under varying degrees of distribution shift. A higher degree indicates a larger distribution shift; (b). **Effects of Sample Size**. OOD generalization versus number of samples used in fine-tuning stage; (c). **Effects of Fine-tuning LR**. OOD Generalization versus fine-tuning learning rates for models ContextPred, AttrMask, and Mole-BERT on the Drug-OOD dataset. (d). **Relation to ID Performance.** OOD versus ID performances (measured by ROC-AUC) of three pre-trained models on Drug-OOD and MoleculeNet datasets.

**Effect of the Distribution Shift Degrees.** We investigate the relationship between the performance drop and shift degrees. To quantify shift degrees, we adopt the following approach: First, we train a vanilla GNN model on the training domain without considering distribution shift. Subsequently, we evaluate the performance drop on the testing domain with distribution shifts. Specifically, we calculate the relative performance drop in AUC-ROC for multiple seeds and use the average value to represent the shift degree. The formula for calculating shift degree ($\Delta S$) is given by:

$$\Delta S = \frac{1}{n} \sum_{i=1}^{n} \left( \frac{\text{AUC-ROC}_{\text{train}} - \text{AUC-ROC}_{\text{test},\,i}}{\text{AUC-ROC}_{\text{train}}} \right) \tag{4}$$

where $\text{AUC-ROC}_{\text{train}}$ is the AUC-ROC score achieved by the GNN model on the training domain without distribution shift, $\text{AUC-ROC}_{\text{test},\,i}$ is the AUC-ROC score achieved by the GNN model on the testing domain with distribution shift for the $i^{th}$ seed, and $n$ is the total number of seeds used. The shift magnitude, $\Delta S$, represents the average relative performance drop across different seeds. Fig. 2(a) illustrates the relationship between performance degradation and the degree of distribution shift on the Drug-OOD dataset, where there is the distribution shift on size. Here $n = 10$. It is evident that a negative correlation exists between performance and shift degrees across all examined methods. Notably, pre-trained models maintain superior performance relative to other methods at all degrees of shift, underscoring their robustness against varying degrees of distribution shifts.

**Effect of the Fine-tuning Sample Size.** We also study the importance of fine-tuning sample size. We test the OOD generlization with {5%, 10%, 20%, 40%, 50%, 65%, 80%} of the size we used in original settings on Drug-OOD and MoleculeNet datasets. Results on Drug-OOD datasets are given

in Fig. 2(b), showing that more samples during fine-tuning lead to better generalization. However, even with only a few samples, pre-trained models still achieve good generalization performance. For instance, with only 20% of the original sample size, the pre-trained models can achieve comparable performances compared with baselines (baseline results are in Table 2).

**Effect of the Fine-tuning Learning Rates.** Based on the theoretical and empirical conclusions drawn from prior work in Euclidean space data (Li et al., 2019; Yu et al., 2021), we explore whether the choice of learning rate during the fine-tuning phase has a consistent impact on OOD generalization. To analyze this relationship, we experimented with a set of learning rates for all pre-trained models, specifically: {0.02, 0.01, 0.005, 0.002, 0.001, 0.0005, 0.0002, 0.0001}. The number of epochs are 100 for all cases. Our empirical investigation shows that models fine-tuned with smaller learning rates achieve better generalization capabilities. Fig. 2(c) gives the OOD generalization performance versus the selection of learning rate for Context prediction, attribute masking and Mole-BERT on Drug-OOD dataset. The results indicate that, only for Mole-BERT, a smaller fine-tune learning rate leads to better generalization performance. While for Attraibute masking and context prediction, there is no correlation between generalization performance and fine-tuning learning rates, which contrary to the findings in image data (Yu et al., 2021).

**Relation to the In-distribution Performance.** In considering the relevance of pre-trained models to downstream tasks, a question arises: Is the inherent model capability (shown as the ID learning performances), reflected by the model's performance on its pre-training dataset, crucial for OOD generalization in downstream tasks? To analyze this association, we evaluated the relationship between the generalization performances with OOD and in-distribution (ID) learning on Drug-OOD and MoleculeNet datasets. Specifically, ID performances are the down-streaming generalizaiton results of the pre-trained models (pre-trained on ZINC-15 dataset) on Drug-OOD and MoleculeNet datasets without ditsirbution shift. Fig. 2(d) gives the evaluation, indicating that there is no clear correlation between OOD and ID performances. This finding shows that "accuracy on the line" phenomenon (Miller et al., 2021) does not always hold for the graph pre-trained models under OOD generlization problem.

## 5 CONCLUSIONS

Our work is placed within a context where prior methods have designed relatively complicated algorithms tailored for Graph OOD. It is crucial to clarify that our research does not challenge or discredit these existing methods; instead, we offer the perspective by evaluating and benchmarking the performance of pre-trained models on Graph OOD problems.

**The Potential of Pre-trained Models for Graph OOD:** We discovered that various pre-trained models, with minimal fine-tuning, could match and often surpass, the performance of methods specially for graph OOD, such as invariant/causal learning and data augmentation. This is especially evident in tasks involving molecular graphs, regardless of the type of distribution shift (concept or covariate), where the pre-trained models achieved superior OOD generalization compared to baseline methods in most cases. Significantly, our results demonstrate that pre-trained models are consistently well-performing among all distribution shift degrees, showing the advantages in OOD scenarios.

**In-depth Empirical Study on Pre-trained Models for Graph OOD:** Our empirical investigation seeks to provide a deeper understanding of the role of the pre-trained models and various design choices for fine-tuning play in ensuring optimal OOD generalization. Specifically, we explored the correlation between fine-tuning learning rate and OOD generalization, the relationship between pre-trained models in OOD and ID scenarios, and the impact of sample size, providing empirical insights that can guide future research in OOD and pre-training.

In future work, we aim to explore a broader range of pre-training methods and OOD scenarios, building upon the current evaluation of representative approaches. The development of model selection strategies, particularly in the context of pre-trained models and OOD generalization, is identified as a promising avenue. Additionally, the potential enhancement of OOD generalization performance through the combination of pre-trained models with invariant learning or data augmentation techniques is suggested. The exploration of theoretical connections between graph pre-training and OOD, drawing inspiration from self-supervised learning and pre-train models, is also highlighted as a direction for further investigation. The detailed discussion on furture directions is given in Appendix A.

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

# SUPPLEMENTARY MATERIAL OF
# THE UNREASONABLE EFFECTIVENESS OF PRETRAINING IN GRAPH OOD

## TABLE OF CONTENTS

## A  DETAILED DISCUSSION ON FUTURE WORK

### A.1  EXPLORATION OF MORE PRE-TRAINING METHODS AND OOD SCENARIOS

Our current work predominantly evaluates representative pre-training and OOD methods/scenarios. However, the field abounds with numerous other methodologies, as summarized in several surveys (Li et al., 2022c; Xia et al., 2022). Due to computational constraints, we could not explore each one exhaustively, leaving a potential avenue for future research.

### A.2  DEVELOPMENT OF MODEL SELECTION APPROACHES

Our empirical evaluations, especially those concerning learning rate experiments, lead us to believe that developing pre-trained model selection strategies (e.g., (You et al., 2022)) for OOD generalization is a promising direction for future research.

### A.3 COMBINATION OF METHODS FOR ENHANCED PERFORMANCE

Future studies could potentially combine pre-trained models with invariant learning or data augmentation techniques to attain improved OOD generalization performance.

### A.4 POTENTIAL THEORETICAL UNDERSTANDING

Based on our current evaluations, there exists an opportunity to explore theoretical connections between graph pre-training and OOD, providing a richer, more in-depth understanding of the empirical performance. One potential direction is exploring some theoretical findings in self-supervised learning and pre-train models (Lee et al., 2021).

## B DETAILS ON DATASETS

### B.1 DATASET STATISTICS

Table 1 summarizes the important key factors and statistics of the molecular datasets. Table A1 and A2 give the full dataset and graph statistics of molecular and general graph datasets used in the paper, respectively.

Table A1: Split statistics of general graph datasets.

| Datasets | Domain | Shift | #. Graphs (training/validation/testing) | Avg. #. Node (training/validation/testing) | Avg. #. Edge (training/validation/testing) | #. Classes | Metrics |
|---|---|---|---|---|---|---|---|
| Motif | Basis | Covariate | 18,000/3,000/3,000 | 17.1/15.8/14.9 | 48.9/33.0/31.5 | 3 | Accuracy |
| | | Concept | 12,600/6,000/6,000 | 16.9/17.0/17.0 | 48.5/48.9/48.7 | | |
| | Size | Covariate | 18,000/3,000/3,000 | 16.9/39.2/87.2 | 43.6/107.0/239.6 | | |
| | | Concept | 12,600/6,000/6,000 | 51.8/51.5/51.6 | 141.8/140.2/141.5 | | |
| CMNIST | Color | Covariate | 42,000/7,000/7,000 | 75.0/75.0/75.0 | 1392.8/1393.7/1392.6 | 10 | Accuracy |
| | | Concept | 29,400/14,000/14,000 | 75.0/75.0/75.0 | 1392.8/1393.5/1392.9 | | |

Table A2: Split statistics of molecular datasets.

| Datasets | Domain | Shift | #. Graphs (training/validation/testing) | Avg. #. Node (training/validation/testing) | Avg. #. Edge (training/validation/testing) | #. Classes / Task | #. Task | Metrics |
|---|---|---|---|---|---|---|---|---|
| DrugOOD | Scaffold | | 21,519/19,041/19,048 | 39.4/26.8/22.5 | 85.8/58.4/47.7 | 2 | | |
| | Assay | | 34,179/19,028/19,032 | 34.5/30.7/29.7 | 75.2/66.8/64.7 | 2 | 1 | |
| | Size | | 36,597/17,660/16,415 | 38.0/25.6/20.0 | 82.8/56.0/43.3 | 2 | | |
| BBBP | | | 1,631/204/204 | 22.5/33.4/27.5 | 48.4/72.3/59.8 | 2 | 1 | |
| Tox21 | | | 6,264/783/784 | 16.5/26.8/26.6 | 33.7/58.1/57.8 | 2 | 12 | |
| ToxCast | | Covariate | 6,860/858/858 | 16.7/26.2/28.2 | 33.5/56.2/60.8 | 2 | 617 | |
| SIDER | Scaffold | | 1,141/143/143 | 30.0/43.2/53.3 | 62.8/91.8/112.7 | 2 | 27 | ROC-AUC |
| ClinTox | | | 1,181/148/148 | 25.5/32.6/24.6 | 54.2/71.0/53.4 | 2 | 2 | |
| MUV | | | 74,469/9,309/9,309 | 24.0/25.3/25.3 | 51.8/55.6/55.5 | 2 | 17 | |
| HIV | | | 32,901/4,113/4,113 | 25.3/27.8/25.3 | 54.1/61.1/55.6 | 2 | 1 | |
| BACE | | | 1,210/151/152 | 33.6/37.2/34.8 | 72.6/81.3/75.1 | 2 | 1 | |
| OGBG-MolHIV | Scaffold | Covariate | 24,682/4,113/4,108 | 26.2/24.9/19.8 | 56.7/54.5/40.6 | 2 | 1 | |
| | | Concept | 15,274/9,382/9,927 | 24.6/26.5/26.6 | 53.1/56.9/57.1 | | | |
| | Size | Covariate | 26,169/2,773/3,961 | 27.8/15.5/12.1 | 60.1/32.8/24.9 | | | |
| | | Concept | 14,483/9,676/10,762 | 31.3/20.0/19.4 | 67.7/42.8/41.5 | | | |
| OGBG-MolPCBA | Scaffold | Covariate | 262,764/44,019/43,562 | 26.9/23.7/20.9 | 58.2/51.6/44.6 | 2 | 128 | AP |
| | | Concept | 159,158/90,740/119,821 | 25.5/26.4/26.7 | 55.2/57.0/57.7 | | | |
| | Size | Covariate | 269,990/48,430/31,925 | 27.9/19.1/15.0 | 60.5/40.9/31.5 | | | |
| | | Concept | 150,121/108,267/115,205 | 27.6/24.5/24.4 | 59.8/53.0/52.6 | | | |
| NCI1 | | | 1,942/215/412 | 20.8/20.7/61.1 | 44.6/44.6/132.9 | 2 | 1 | |
| NCI109 | Size | Covariate | 1,872/207/421 | 20.4/20.3/61.1 | 43.8/43.6/133.1 | 2 | 1 | MCC |
| PROTEINS | | | 511/56/112 | 15.4/15.7/138.9 | 57.4/58.5/504.6 | 2 | 1 | |
| DD | | | 533/59/118 | 143.2/156.1/746.4 | 707.1/746.4/3814.7 | 2 | 1 | |

### B.2 DETAILS ON DATASET INTRODUCTION

**DrugOOD** (Ji et al., 2023). This benchmark supports AI-driven drug discovery with realistic molecular graph datasets. It automates OOD dataset curation using ChEMBL (Mendez et al., 2019) and offers diverse dataset splitting criteria, including scaffold, assay type and size, for tailored domain alignment. The task focus on drug target binding affinity prediction.

**MoleculeNet** (Wu et al., 2018). MoleculeNet stands as a comprehensive benchmark for molecular machine learning. It curates diverse public datasets, sets up evaluation standards, and offers open-source tools for different molecular learning methods, all accessible via the DeepChem open source library (Ramsundar et al., 2019).

The benchmark comprises multiple binary graph classification datasets, each designed to evaluate model performance across different facets of molecular interaction. Specifically, BBBP (Martins et al., 2012) evaluates the crucial measure of blood-brain barrier penetration, vital for understanding membrane permeability. Tox21 (Abdelaziz et al., 2016) offers toxicity data encompassing 12 biological targets, including nuclear receptors and stress response pathways. Toxcast (Richard et al., 2016) provides toxicology measurements based on over 600 in vitro high-throughput screenings, serving as a rich resource for understanding toxicity. SIDER (Kuhn et al., 2016) features a database detailing marketed drugs and adverse drug reactions, categorized into 27 system organ classes, offering insights into drug safety. ClinTox (Novick et al., 2013) (AAC) consists of qualitative data classifying drugs approved by the FDA and those that have failed clinical trials due to toxicity concerns. MUV (Gardiner et al., 2011) represents a subset of PubChem BioAssay (Kim et al., 2023), refined through nearest neighbor analysis, and tailored for validating virtual screening techniques. The HIV dataset originates from the Drug Therapeutics Program (DTP) AIDS Antiviral Screen (Riesen & Bunke, 2008), a comprehensive screening effort that evaluated the effectiveness of more than 40,000 compounds in inhibiting HIV replication. BACE (Subramanian et al., 2016) is a dataset that provides qualitative binding results for a collection of inhibitors targeting human $\beta$-secretase 1.

**OGBG** (Hu et al., 2020). OGBG is a specific subset within Open Graph Benchmark (OGB), containing representative datasets like OGBG-Molhiv, OGBG-Molpcba, and OGBG-PPA. OGBG-Molhiv and OGBG-Molpcba challenge graph property prediction with distribution shifts, specifically focusing on predicting molecular properties. They use a scaffold splitting approach, separating structurally distinct molecules into different subsets for a realistic evaluation of graph generalization. The dataset split follows GOOD benchmark (Gui et al., 2022). Specifically, for covariate shift with a distribution source of size, we arranged the molecules in descending order based on the number of nodes and split them into a ratio of $8 : 1 : 1$ for the training set, validation set, and testing set, respectively. Similarly, the entire dataset was ordered based on the Bemis-Murcko scaffold string of SMILES, maintaining the same ratio. For concept shift, exemplified by size, we categorized molecules into different groups based on different numbers of molecular nodes. Following this categorization, we selected samples from each group with different labels, forming the training set, validation set, and testing set, respectively, with a ratio of $3 : 1 : 1$. This grouping approach aligns with the scaffold-wise distribution, where molecules are categorized based on the Bemis-Murcko scaffold string of SMILES.

**TU Datasets.** (Morris et al., 2020) It is a collection of benchmark datasets for graph classification and regression. Among these datasets, NCI1, NCI109, PROTEINS, and DD stand out as important and representative graph classification datasets, each offering unique characteristics and complexities. NCI1 and NCI109 datasets are prominent in chemoinformatics. NCI1 is a binary graph classification dataset that focuses on anticancer compound classification. It comprises molecular graphs, with nodes representing atoms and edges indicating chemical bonds. NCI109 extends the challenge by expanding the number of classes and compounds. PROTEINS is a dataset focused on protein graphs, where each node represents a specific protein, and the edges signify various biologically relevant connections or associations between these proteins. The task is to predict the presence or absence of specific protein functions. DD is a real-world graph classification dataset, comprising $1,178$ protein network structures, each of which features 82 distinct node labels. The task is to classify each graph into one of two classes: an enzyme or a non-enzyme.

**Motif.** Motif is a synthetic dataset (Wu et al., 2022b). It has been created to address structural shifts in graph data. In this dataset, each graph is composed of a base and a motif. The bases are categorized into three distinct types: Tree ($S = 0$), Ladder ($S = 1$), and Wheel ($S = 2$). On the other hand, the motifs include Cycle ($C = 0$), House ($C = 1$), and Crane ($C = 2$), introducing various structural complexities into the dataset. The ground truth label $Y$ for each graph is exclusively dictated by the motif ($C$). The primary objective in this dataset is to accurately classify the graphs into one of three classes: Cycle, House, or Crane.

**CMNIST.** CMNIST is a special dataset with graphs showcasing handwritten digits. These graphs are created from the MNIST dataset (Arjovsky et al., 2019) but preprocessed with superpixel (Monti et al., 2017). The goal is to classify each graph into one of the ten-digit categories, from 0 to 9.

# C DETAILS ON EVALUATED METHODOLOGIES

Fig. A1 gives the evaluation pipeline on pre-trained GNNs for graph OOD secenarios with a showing case on molecular graphs.

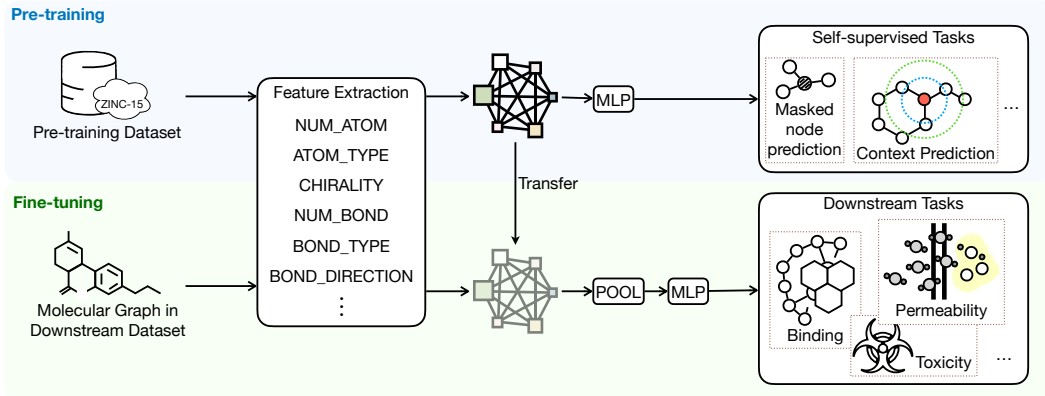

Figure A1: PodGenGraph pipeline for molecular graph pre-training and fine-tuning for downstream datasets.

## C.1 HYPERPARAMETER DETAILS FOR BASELINE METHODS

**CIGA**. We used default hyperparameters as specified in the original paper for DrugOOD, TU datasets, Motif, and CMNIST. Specifically, in DrugOOD, the causal substructure size is set to 80% of each graph size for DrugOOD-Scaffold and DrugOOD-Assay, while it's 10% for DrugOOD-Size. The dropout rate is 0.5 for DrugOOD-Scaffold and DrugOOD-Assay, and 0.1 for DrugOOD-Size. For DrugOOD-Assay with CIGA-v1 and CIGA-v2, the coefficient for contrastive loss is set to 8 and 1, respectively. For DrugOOD-Scaffold with CIGA-v1 and CIGA-v2, it's 32 and 16, respectively. For DrugOOD-Size with CIGA-v1 and CIGA-v2, it's 16 and 2, respectively.

For TU datasets, we use a causal substructure size of 60% for NCI1, 70% for NCI109, and 30% for DD and PROTEINS. The coefficient for contrastive loss is 0.5 for NCI1 with CIGA-v1 and 1 for NCI1 with CIGA-v2. It's 2 for both NCI109 and DD with all CIGA versions. For PROTEINS, the coefficient for contrastive loss is 0.5 with both CIGA-v1 and CIGA-v2.

In Motif, the causal substructure ratio is 25%, and in CMNIST, it's 80%. For Motif, the coefficient of contrastive loss is chosen from {0.5, 1, 4, 8, 16, 32}, and for CMNIST, it's 32 with CIGA-v1 and 16 with CIGA-v2.

For datasets in MoleculeNet and scaffold distribution shift in OGBG datasets, we use hyperparameters similar to those in DrugOOD-Scaffold. For size distribution shift in OGBG datasets, the hyperparameters are aligned with those in DrugOOD-Size.

**MoleOOD**. We employed default hyperparameters as provided in the code release. Specifically, we selected the prior distribution from uniform, Gaussian distribution for all datasets. In DrugOOD, we utilized 20 domains for the domain prior across three datasets. For MoleculeNet and OGBG datasets, we varied the number of domains among {10, 15, 20}.

**LiSA**. We utilized the default hyperparameters provided in the code release. The inner loop was set to 20 for all datasets. We employed 3 subgraph generators and a coefficient loss regularization term of 0.1 across all datasets.

# D   FULL RESULTS

## D.1   RESULTS ON DIFFERENT DATASETS.

Appendix Table A3-A9 give the full results on the OOD performances of all evaluated methods sperated by datasets.

Table A3: Testing ROC-AUC on Drug-OOD datasets (Ji et al., 2023) with covariate shift. Blue shaded rows indicate pre-training strategies. The first and second best-performing methods (except the ID training) are in **bold** and **bold**, respectively.

| | DrugOOD-Scaffold | DrugOOD-Assay | DrugOOD-Size | Avg |
|---|---|---|---|---|
| CIGA-v1 | $69.27_{\pm0.81}$ | $72.36_{\pm0.60}$ | $67.08_{\pm0.82}$ | 69.57 |
| CIGA-v2 | $69.68_{\pm0.21}$ | $\mathbf{73.28_{\pm0.35}}$ | $68.02_{\pm0.51}$ | $\underline{\mathbf{70.32}}$ |
| MoleOOD | $68.01_{\pm0.39}$ | $71.18_{\pm0.63}$ | $66.61_{\pm0.36}$ | 68.60 |
| LiSA | $65.71_{\pm0.25}$ | $67.66_{\pm0.63}$ | $65.78_{\pm0.46}$ | 66.38 |
| ContextPred | $70.01_{\pm0.13}$ | $\underline{\mathbf{72.80_{\pm0.55}}}$ | $\mathbf{68.42_{\pm0.10}}$ | $\mathbf{70.41}$ |
| AttrMask | $\mathbf{70.68_{\pm0.31}}$ | $71.56_{\pm0.43}$ | $\underline{\mathbf{68.22_{\pm0.15}}}$ | 70.15 |
| Mole-BERT | $\underline{\mathbf{70.04_{\pm0.25}}}$ | $71.19_{\pm0.09}$ | $67.92_{\pm0.19}$ | 69.60 |
| GIN-OOD | $67.31_{\pm0.50}$ | $71.20_{\pm0.29}$ | $66.67_{\pm0.26}$ | 68.39 |
| GIN-ID | $84.36_{\pm0.15}$ | $87.07_{\pm0.62}$ | $87.69_{\pm0.77}$ | 86.37 |

Table A4: Testing ROC-AUC on MoleculeNet datasets (Wu et al., 2018) with covariate shift. Blue shaded rows indicate pre-training strategies.

| | BBBP | Tox21 | ToxCast | SIDER | ClinTox | MUV | HIV | BACE | Avg |
|---|---|---|---|---|---|---|---|---|---|
| CIGA-v1 | $65.50_{\pm1.62}$ | $73.87_{\pm0.54}$ | $62.81_{\pm0.55}$ | $57.40_{\pm4.40}$ | $55.00_{\pm1.60}$ | $68.10_{\pm1.30}$ | $75.79_{\pm1.09}$ | $73.60_{\pm4.30}$ | 67.75 |
| CIGA-v2 | $68.69_{\pm1.37}$ | $72.25_{\pm1.46}$ | $58.53_{\pm1.85}$ | $54.90_{\pm2.13}$ | $66.37_{\pm3.22}$ | $70.99_{\pm1.34}$ | $73.19_{\pm4.22}$ | $78.56_{\pm2.34}$ | 68.16 |
| MoleOOD | $\underline{\mathbf{69.71_{\pm1.56}}}$ | $73.65_{\pm0.85}$ | $62.90_{\pm0.96}$ | $\mathbf{62.01_{\pm0.58}}$ | $\mathbf{89.93_{\pm3.90}}$ | $67.79_{\pm2.46}$ | $\mathbf{78.29_{\pm0.51}}$ | $\mathbf{81.10_{\pm1.97}}$ | $\underline{\mathbf{73.36}}$ |
| LiSA | $65.26_{\pm2.01}$ | $66.32_{\pm0.76}$ | $59.56_{\pm0.57}$ | $57.28_{\pm0.66}$ | $65.00_{\pm2.60}$ | $67.91_{\pm1.13}$ | $62.57_{\pm1.30}$ | $69.97_{\pm3.06}$ | 64.92 |
| ContextPred | $69.32_{\pm1.03}$ | $74.47_{\pm0.36}$ | $\underline{\mathbf{63.43_{\pm0.40}}}$ | $60.45_{\pm0.60}$ | $57.40_{\pm3.16}$ | $\underline{\mathbf{77.36_{\pm1.11}}}$ | $77.56_{\pm0.95}$ | $79.41_{\pm1.96}$ | 68.38 |
| AttrMask | $64.95_{\pm3.40}$ | $\mathbf{76.22_{\pm0.41}}$ | $63.36_{\pm0.50}$ | $60.15_{\pm0.57}$ | $70.47_{\pm3.43}$ | $74.93_{\pm2.07}$ | $76.41_{\pm0.70}$ | $\underline{\mathbf{79.88_{\pm0.61}}}$ | 71.37 |
| Mole-BERT | $\mathbf{71.88_{\pm1.12}}$ | $\underline{\mathbf{76.90_{\pm0.33}}}$ | $\mathbf{64.18_{\pm0.31}}$ | $\underline{\mathbf{62.74_{\pm0.89}}}$ | $\underline{\mathbf{78.88_{\pm2.24}}}$ | $\mathbf{78.62_{\pm1.51}}$ | $\underline{\mathbf{78.10_{\pm0.65}}}$ | $80.88_{\pm1.45}$ | $\mathbf{74.62}$ |
| GIN-OOD | $65.78_{\pm4.90}$ | $73.95_{\pm0.28}$ | $62.13_{\pm0.71}$ | $57.38_{\pm1.65}$ | $57.29_{\pm5.91}$ | $70.40_{\pm1.80}$ | $75.06_{\pm2.06}$ | $70.78_{\pm5.29}$ | 66.70 |
| GIN-ID | $93.13_{\pm0.58}$ | $82.60_{\pm0.20}$ | $70.93_{\pm0.28}$ | $62.57_{\pm0.81}$ | $84.91_{\pm2.10}$ | $79.49_{\pm1.44}$ | $80.86_{\pm1.11}$ | $86.73_{\pm1.72}$ | 80.55 |

Table A5: Performance evaluation on OGBG datasets (Hu et al., 2020) with covariate shift. OGBG-MolPCBA is evaluated by AP, while OGBG-MolHIV is evaluated by ROC-AUC. Blue shaded rows indicate pre-training strategies. The first and second best-performing methods (except the ID training) are in **bold** and **bold**, respectively.

| | OGBG-MolPCBA | | OGBG-MolHIV | |
|---|---|---|---|---|
| | Size | Scafflod | Size | Scafflod |
| CIGA-v1 | $10.51_{\pm0.17}$ | $10.24_{\pm1.98}$ | $61.81_{\pm1.68}$ | $69.40_{\pm2.39}$ |
| CIGA-v2 | $9.65_{\pm0.12}$ | $10.62_{\pm1.04}$ | $59.55_{\pm2.56}$ | $69.40_{\pm1.97}$ |
| LiSA | $6.52_{\pm0.20}$ | $8.67_{\pm0.24}$ | $59.65_{\pm1.44}$ | $68.92_{\pm0.92}$ |
| ContextPred | $13.30_{\pm0.37}$ | $\mathbf{22.14_{\pm0.43}}$ | $60.47_{\pm0.88}$ | $\mathbf{70.69_{\pm1.12}}$ |
| AttrMask | $\underline{\mathbf{13.50_{\pm0.38}}}$ | $\underline{\mathbf{21.89_{\pm0.27}}}$ | $\underline{\mathbf{62.29_{\pm0.91}}}$ | $\underline{\mathbf{70.29_{\pm1.57}}}$ |
| Mole-BERT | $\mathbf{16.19_{\pm0.24}}$ | $17.33_{\pm0.12}$ | $\mathbf{66.95_{\pm0.93}}$ | $69.63_{\pm0.96}$ |
| GIN-OOD | $12.85_{\pm0.34}$ | $13.03_{\pm0.43}$ | $60.06_{\pm1.63}$ | $65.41_{\pm1.70}$ |
| GIN-ID | $28.10_{\pm0.69}$ | $30.80_{\pm0.54}$ | $79.49_{\pm0.55}$ | $80.86_{\pm1.11}$ |

Table A6: Testing Matthews correlation coefficient on TU datasets with covariate shift. Blue shaded rows indicate pre-training strategies. The first and second best-performing numbers (except the ID training) are in **bold** and **bold**, respectively.

| | NCI1 | NCI109 | PROTEINS | DD |
|---|---|---|---|---|
| CIGA-v1 | $0.22_{\pm 0.07}$ | $0.23_{\pm 0.09}$ | $0.40_{\pm 0.06}$ | $0.29_{\pm 0.08}$ |
| CIGA-v2 | $\underline{\mathbf{0.27}}_{\pm 0.07}$ | $0.22_{\pm 0.05}$ | $0.31_{\pm 0.12}$ | $\underline{\mathbf{0.26}}_{\pm 0.08}$ |
| LiSA | $0.24_{\pm 0.01}$ | $0.26_{\pm 0.02}$ | $\underline{\mathbf{0.43}}_{\pm 0.05}$ | $\mathbf{0.37}_{\pm 0.07}$ |
| InfoGraph | $\mathbf{0.39}_{\pm 0.01}$ | $\mathbf{0.38}_{\pm 0.01}$ | $\mathbf{0.53}_{\pm 0.07}$ | $\underline{\mathbf{0.35}}_{\pm 0.04}$ |
| GIN-OOD | $0.21_{\pm 0.06}$ | $0.16_{\pm 0.05}$ | $0.23_{\pm 0.05}$ | $0.25_{\pm 0.09}$ |
| GIN-ID | $0.45_{\pm 0.03}$ | $0.44_{\pm 0.02}$ | $0.46_{\pm 0.03}$ | $0.40_{\pm 0.04}$ |

Table A7: Testing accuracy on general graph datasets with covariate shift. Blue shaded rows indicate pre-training strategies. The first and second best-performing numbers (except the ID training) are in **bold** and **bold**, respectively.

| | Motif | | CMNIST |
|---|---|---|---|
| | Basis | Size | |
| CIGA-v1 | $66.43_{\pm 11.31}$ | $49.14_{\pm 8.34}$ | $\underline{\mathbf{32.22}}_{\pm 2.67}$ |
| CIGA-v2 | $67.15_{\pm 8.19}$ | $\underline{\mathbf{54.42}}_{\pm 3.11}$ | $32.11_{\pm 2.53}$ |
| LiSA | $\underline{\mathbf{82.55}}_{\pm 7.18}$ | $\mathbf{62.90}_{\pm 8.30}$ | $\mathbf{33.21}_{\pm 13.43}$ |
| InfoGraph | $\mathbf{86.85}_{\pm 2.43}$ | $53.43_{\pm 8.09}$ | $24.39_{\pm 2.09}$ |
| GIN-OOD | $62.01_{\pm 3.92}$ | $52.94_{\pm 2.93}$ | $26.28_{\pm 5.95}$ |
| GIN-ID | $92.15_{\pm 0.04}$ | $92.16_{\pm 0.07}$ | $77.80_{\pm 0.20}$ |

## D.2 DIFFERENT STATISTICAL METRICS

Appendix Fig. A2-A3 show the additional statistical evaluation on the performances of all approaches on Drug-OOD and Molecule-Net datasets. The metrics include median, IQM, mean, and the optimality gap. Results also reveal that the pre-trained models achieve well-performance results compared with baseline approaches.

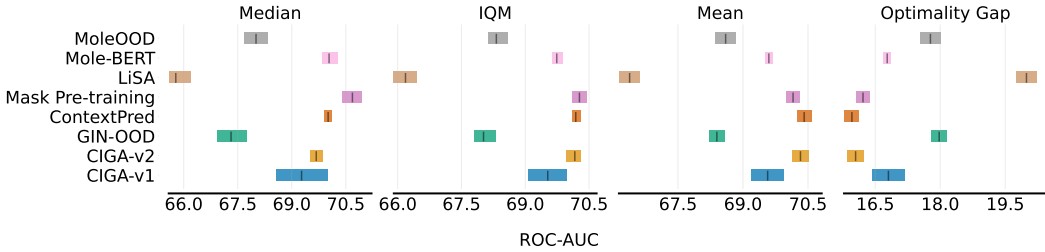

Figure A2: Aggregate performance on DrugOOD averaged across three datasets: `DrugOOD-lbap-core-ic50-assay`, `DrugOOD-lbap-core-ic50-scaffold`, and `DrugOOD-lbap-core-ic50-size`. Better results are indicated by higher mean, median, and IQM scores, along with a lower optimality gap.

Table A8: Performance evaluation on OGBG datasets (Hu et al., 2020) with concept shift. OGBG-MolPCBA is evaluated by AP, while OGBG-MolHIV is evaluated by ROC-AUC. Blue shaded rows indicate pre-training strategies. The first and second best-performing numbers (except the ID training) are in **bold** and **bold**, respectively.

| | OGBG-MolPCBA | | OGBG-HIV | |
| --- | --- | --- | --- | --- |
| | Size | Scafflod | Size | Scafflod |
| CIGA-v1 | $9.22_{\pm0.09}$ | $8.33_{\pm0.06}$ | $72.80_{\pm1.35}$ | $70.79_{\pm1.55}$ |
| CIGA-v2 | $8.31_{\pm0.12}$ | $8.71_{\pm0.12}$ | $\underline{73.62}_{\pm1.33}$ | $\underline{71.65}_{\pm1.33}$ |
| LiSA | $5.05 \pm 0.32$ | $8.55_{\pm0.63}$ | $72.36_{\pm4.75}$ | $69.46_{\pm0.83}$ |
| ContextPred | $11.39_{\pm0.21}$ | $15.71_{\pm0.38}$ | $70.41_{\pm0.38}$ | $68.77_{\pm0.90}$ |
| AttrMask | $\underline{11.87}_{\pm0.24}$ | $\underline{16.14}_{\pm0.49}$ | $70.59_{\pm0.58}$ | $71.50_{\pm0.55}$ |
| Mole-BERT | $\mathbf{15.71}_{\pm0.26}$ | $\mathbf{21.29}_{\pm0.53}$ | $\mathbf{75.94}_{\pm0.91}$ | $\mathbf{76.13}_{\pm0.39}$ |
| GIN-OOD | $12.76_{\pm0.62}$ | $17.27_{\pm0.63}$ | $70.20_{\pm1.12}$ | $62.36_{\pm2.20}$ |
| GIN-ID | $28.10_{\pm0.69}$ | $30.80_{\pm0.54}$ | $79.49_{\pm0.55}$ | $80.86_{\pm1.11}$ |

Table A9: Testing accuracy on general graph datasets with concept shift. Blue shaded rows indicate pre-training strategies. The first and second best-performing numbers (except the ID training) are in **bold** and **bold**, respectively.

| | Motif | | CMNIST |
| --- | --- | --- | --- |
| | basis | size | |
| CIGA-v1 | $72.50_{\pm4.02}$ | $58.63_{\pm6.66}$ | $34.80_{\pm3.33}$ |
| CIGA-v2 | $77.48_{\pm2.54}$ | $\mathbf{70.65}_{\pm4.81}$ | $\mathbf{39.39}_{\pm3.30}$ |
| LiSA | $\mathbf{87.89}_{\pm1.61}$ | $\underline{70.36}_{\pm2.61}$ | $\underline{36.56}_{\pm0.40}$ |
| InfoGraph | $\underline{79.36}_{\pm1.12}$ | $64.79_{\pm1.68}$ | $19.19_{\pm2.17}$ |
| GIN-OOD | $72.12_{\pm1.89}$ | $58.23_{\pm1.73}$ | $29.53_{\pm0.50}$ |
| GIN-ID | $92.15_{\pm0.04}$ | $92.16_{\pm0.07}$ | $77.80_{\pm0.20}$ |

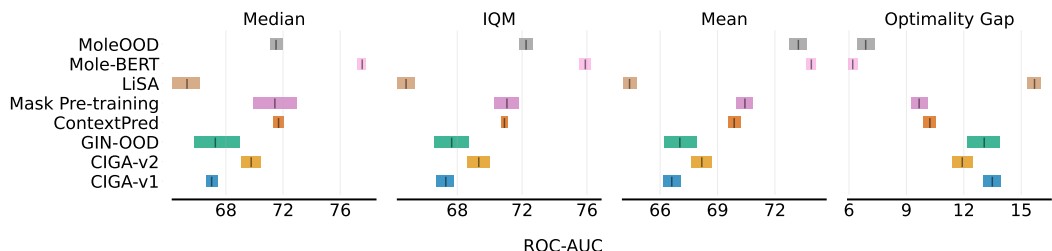

Figure A3: Aggregate performance on MoleculeNet averaged across eight datasets: BBBP, Tox21, ToxCast, SIDER, ClinTox, MUV, HIV, BACE. Better results are indicated by higher mean, median, and IQM scores, along with a lower optimality gap.

## D.3 DIFFERENT BACKBONES

Appendix Fig. A4-A7 show the performance on molecular prediction with different GNN architectures (GIN and GAT).

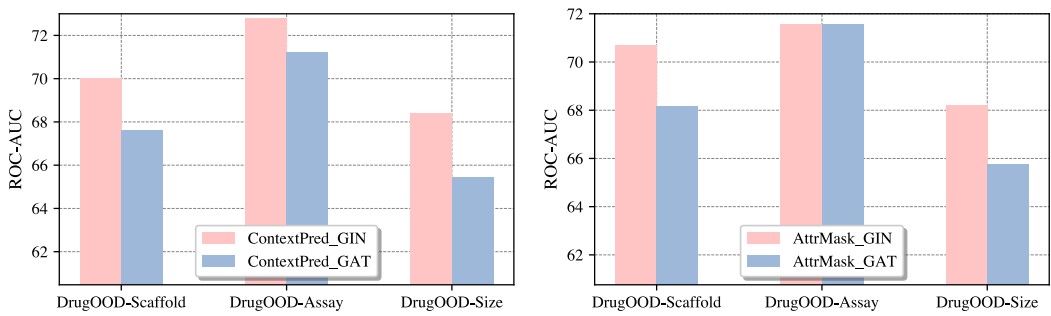

Figure A4: Comparison of ROC-AUC performance (%) on the DrugOOD dataset using the GIN and GAT backbones, respectively.

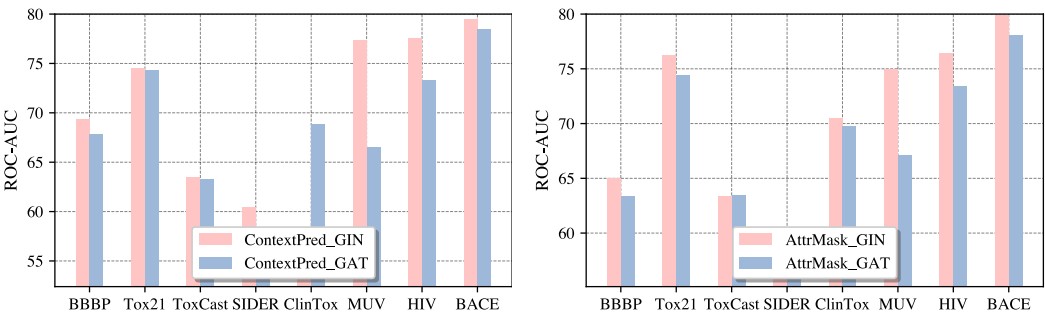

Figure A5: Comparison of ROC-AUC performance (%) on the MoleculeNet dataset using the GIN and GAT backbones, respectively.

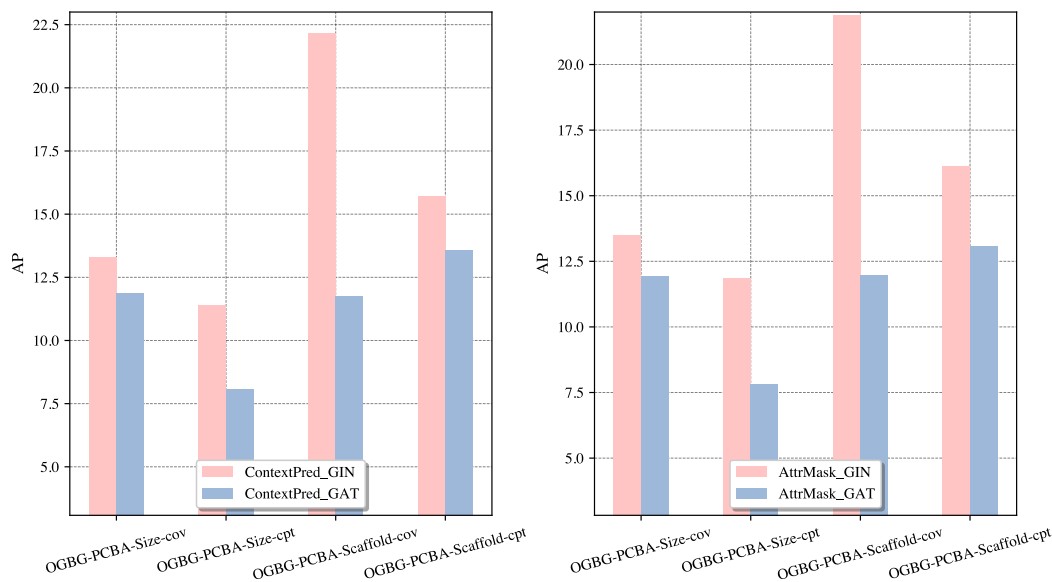

Figure A6: Comparison of AP on the OGBG-PCBA dataset using the GIN and GAT backbones, respectively.

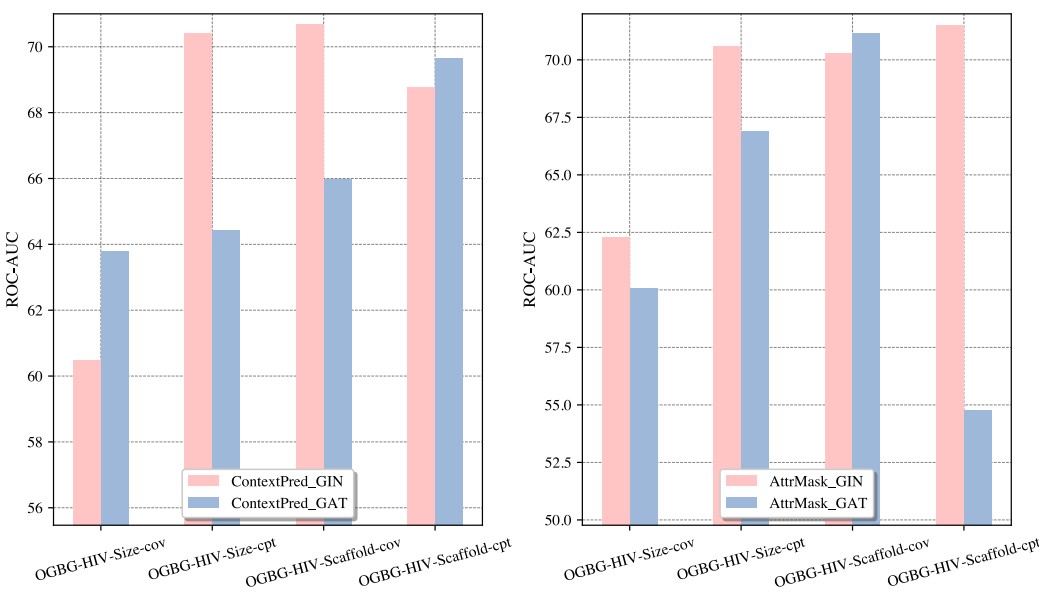

Figure A7: Comparison of ROC-AUC performance (%) on the OGBG-HIV dataset using the GIN and GAT backbones, respectively.

# E REPRODUCIBILTY STATEMENT

## E.1 DETAILS

The experiments are implemented on an 8 Intel Xeon Gold 5220R and 4 NVidia A100 GPUs. We use the publicly accessible code libraries of all evaluated methods. The detailed implementation can be found through this anonymous link: https://sites.google.com/view/podgengraph/.

### E.2 Used Libraries and Licenses

In our implementation, we have used the following libraries which are covered by the corresponding licenses:

- Tensorflow (Apache License 2.0)
- Pytorch (BSD 3-Clause "New" or "Revised" License)
- Numpy (BSD 3-Clause "New" or "Revised" License)
- RDKit (BSD 3-Clause "New" or "Revised" License)
- scikit-image (BSD 3-Clause "New" or "Revised" License)
- wilds (MIT License)
- Codebase of CIGA: link, (MIT license)
- Mole-OOD: link, (MIT license )
- Codebase of LiSA: link
- Codebase of Mask pretraining and context prediction: link, (MIT Liecense)
- Codebase of InfoGraph: link
- Codebase of Molecule-BERT: link

