# OpenReview forum: "The Unreasonable Effectiveness of Pretraining in Graph OOD"
_ICLR.cc/2024/Conference — Submitted to ICLR 2024_

### Official Review · Reviewer_xDkS · 2023-10-30

**Soundness:** 3 good
**Presentation:** 2 fair
**Contribution:** 2 fair
**Rating:** 5
**Confidence:** 4

**Summary:**

The article empirically investigates the potential of graph pre-trained models in handling graph out-of-distribution (OOD) problems. The paper finds that even basic graph pre-trained models often outperform specifically-designed OOD methods for handling distribution shifts. The extensive experiments conducted across various datasets under different distribution shifts scenarios. However, it seems that graph pre-training was originally introduced to address OOD problems. It is unclear about the commonalities and distinctions between graph pre-training and OOD models.

**Strengths:**

The experimental design include a wide range of OOD datasets and distribution shifts scenarios. The phenomenon observed in the experiments can inspire and lead to further exploration.

**Weaknesses:**

1. The commonalities and distinctions between graph pretraining and OOD models should be clearly and comprehensively elucidated. From my perspective, it appears that graph pretraining was originally introduced to address OOD problems. Also in Section 2, the authors include the self-supervised learning into Graph OOD, however some self-supervised learning methods like GraphCL is typically considered as a graph pre-training method.

2. The OOD datasets and pre-trained models used in the article are primarily focused on the molecular domain. Does this imply that the scope defined in the article is mainly centered in molecular graphs? There exist many graph pre-trained models [1-3] beyond just molecules, it would be beneficial for the article to include additional datasets and methods beyond just molecules. It is currently unclear are the findings stated in the paper still hold in these pre-trained models.

3. The findings presented in the article regarding the ability of pre-trained models to address OOD issues seem to be intuitive and insufficient, because pre-trained models are originally designed to address OOD. I recommend the authors to explore more interesting findings, such as identifying which pre-training tasks can aid in specific OOD scenarios. Can any findings help us better guide the design of pre-training models or tasks?


[1] You, Yuning, et al. "Graph contrastive learning with augmentations." NeurIPS 2020.

[2] Hou, Zhenyu, et al. "Graphmae: Self-supervised masked graph autoencoders." KDD 2022.

[3] Qiu, Jiezhong, et al. "Gcc: Graph contrastive coding for graph neural network pre-training." SIGKDD 2020.

**Questions:**

See weakness.

---

> ### Author Response · Authors · 2023-11-18
>
> Thank you for the suggestions and feedback. We have conducted several experiments, and the analyses are included in the updated version. Please refer to the following responses:
>
> **Weaknesses**
> >Q1. The commonalities and distinctions between graph pretraining and OOD models should be clearly and comprehensively elucidated. From my perspective, it appears that graph pretraining was originally introduced to address OOD problems. Also in Section 2, the authors include the self-supervised learning into Graph OOD, however some self-supervised learning methods like GraphCL is typically considered as a graph pre-training method.
>
>
> While the initial assumption might suggest that graph pretraining was primarily introduced to address OOD problems, our investigation reveals a more impactful insight. Although the use of pre-training to enhance OOD performance seems intuitive, our findings challenge conventional expectations. Specifically, in the context of molecular graphs, pre-trained models such as Mole-BERT not only demonstrate strong performance in OOD scenarios but, notably, surpass many state-of-the-art methods explicitly designed for OOD tasks. This discovery disrupts the traditional belief that specialized methods would naturally excel in their tailored domains.
>
> We have updated Section 2 with GraphCL accordingly.
>
> >Q2. The OOD datasets and pre-trained models used in the article are primarily focused on the molecular domain. Does this imply that the scope defined in the article is mainly centered in molecular graphs? There exist many graph pre-trained models [1-3] beyond just molecules, it would be beneficial for the article to include additional datasets and methods beyond just molecules. It is currently unclear are the findings stated in the paper still hold in these pre-trained models.
>
> The scope we consider in the paper includes both molecular graphs (DrugOOD, MoleculeNet, OGBG, TU datasets) and simulated general graphs (motifs and CMNIST). We have conducted the evaluation on [You et al., 2020] and [Hou et al., 2022] on DrugOOD, MoleculeNet, and OGBG datasets. See the updated results in Table 2.
>
> >Q3. The findings presented in the article regarding the ability of pre-trained models to address OOD issues seem to be intuitive and insufficient, because pre-trained models are originally designed to address OOD. I recommend the authors to explore more interesting findings, such as identifying which pre-training tasks can aid in specific OOD scenarios. Can any findings help us better guide the design of pre-training models or tasks?
>
> It's crucial to clarify that while the use of pre-training to enhance OOD performance might seem intuitive, our research uncovers a less obvious and more impactful insight. We demonstrate that in terms of molecular graphs, pre-trained models like Mole-BERT not only perform well in OOD scenarios but, quite importantly, outperform many state-of-the-art methods specifically designed for OOD tasks. This finding disrupts the conventional expectation that specialized methods would naturally excel in their tailored domains. Our meta-analysis further delves into variables such as learning rate and sample efficiency, as well as the relation between OOD and in-distribution learning.
>
> [You et al., 2020] You, Yuning, et al. "Graph contrastive learning with augmentations." NeurIPS 2020.
>
> [Hou et al., 2022] Hou, Zhenyu, et al. "Graphmae: Self-supervised masked graph autoencoders." KDD 2022.

---

> > ### Comment · Reviewer_xDkS · 2023-11-23
> > **Thanks for your response**
> >
> > Thank you for the author's response. The experiment of the graph pre-training model still remains limited to the scope of molecular graphs:  many pretraining models are tailored for molecular graphs, so their superior performance in OOD scenarios for molecules seems to be intuitive. I maintain my original score.

---

### Official Review · Reviewer_qBm3 · 2023-10-31

**Soundness:** 2 fair
**Presentation:** 3 good
**Contribution:** 2 fair
**Rating:** 5
**Confidence:** 4

**Summary:**

This paper serves as a benchmark for discussing the use of pre-trained methods in handling Graph-level OOD (Out-of-Distribution) problems. Through experiments conducted on diverse datasets of varying sizes and types of OOD scenarios, the paper finds that models learned through pre-training achieve results in OOD scenarios that are comparable to, or even better than, specialized methods designed for OOD problems. The authors also make meaningful explorations into the key factors influencing the ability of pre-trained models to handle OOD problems.

**Strengths:**

- The motivation behind this paper is well-founded. With the increasing research focus on Graph OOD (Out-of-Distribution) problems, providing a unified and fair benchmark for various methods is highly meaningful.
- The paper's selection of numerous datasets of various types and sizes allows for a comprehensive representation of different methods' strengths and weaknesses across different scenarios.

**Weaknesses:**

- I am uncertain whether 'PODGenGraph' can truly be called a benchmark. As per my understanding, the purpose of a benchmark should be to provide a fair and extensive comparison of the performance of different methods on the same task within the same environment. Since this paper claims to be a benchmark for Pretraining methods in OOD tasks, it should cover a more representative set of pretraining methods. The selection of only three methods in the paper seems limited and unrepresentative (for example, the omission of popular contrastive methods and masked autoencoders).
- The author's claims about the superiority of pretraining methods in Graph OOD tasks appear to be overstated. The author suggests that basic pretraining methods can achieve comparable performance to specially designed methods. However, according to Table 2, there is still a significant performance gap between pretraining methods and specially designed methods on many datasets.
- It's not surprising that pretraining on a large molecular dataset can enhance a model's performance in OOD scenarios due to the semantic similarities within molecular datasets, similar to the way large language models operate. However, as seen in Table 2, pretraining on molecular graphs does not seem to generalize well to the CMNIST dataset. This highlights the limitations of pretraining methods
- The analysis of factors influencing the performance of Pretraining methods in Section 4.3 appears somewhat shallow. For instance, the learning rate does not seem to be a factor worthy of analysis because it is not a critical component of the model. On the other hand, the author claims that pretrained models have a sample-efficient advantage. I initially thought this meant pretrained models can maintain good performance with smaller training sets. However, Figure 2b only demonstrates that pretrained models outperform baseline methods at different label rates.

**Questions:**

- This paper focuses merely on graph-level tasks. I wonder if pretrained methods can yield similar results on node-level OOD tasks.


Though I have raised several points in weaknesses. I am glad to adjust my rating if the reviewer can address my concerns.

---

> ### Author Response · Authors · 2023-11-18
>
> Thank you for your insightful feedback, addressing it will help us make the paper better. In response to your questions, please refer to the following items:
>
> **Weakness**
> >Q1. I am uncertain whether 'PODGenGraph' can truly be called a benchmark. As per my understanding, the purpose of a benchmark should be to provide a fair and extensive comparison of the performance of different methods on the same task within the same environment. Since this paper claims to be a benchmark for Pretraining methods in OOD tasks, it should cover a more representative set of pretraining methods. The selection of only three methods in the paper seems limited and unrepresentative (for example, the omission of popular contrastive methods and masked autoencoders).
>
> We have added  two additional pre-training methods: GraphCL (You et al., 2020) and GraphMAE (Hou et al., 2022), shown in Table 2. This expansion ensures that PODGenGraph covers most of the representative graph pre-trained models currently prevalent in the field. In the paper, we have also evaluated these models across the majority of benchmark datasets and setups pertinent to graph OOD tasks. So we believe this ensures the fair and extensive comparison of different pre-trained graph models under OOD settings.
>
> >Q2. The author's claims about the superiority of pretraining methods in Graph OOD tasks appear to be overstated. The author suggests that basic pretraining methods can achieve comparable performance to specially designed methods. However, according to Table 2, there is still a significant performance gap between pretraining methods and specially designed methods on many datasets.
>
> In molecular datasets, our findings demonstrate that pre-trained models achieve the best or second-best performances across all cases, as detailed in Table 2. We acknowledge that in simpler and simulated datasets like Motif and CMNIST, pre-trained models did not show strong results. This can be attributed to the nature of these tasks where invariant learning through disentangled or causal learning can effectively learn good representations. In such scenarios, the additional benefits brought by pre-trained methods may not be as pronounced, which is a reasonable observation.
>
> However, our primary focus is on addressing real-world graph applications, especially in contexts like molecular data, where labeled data is often scarce during inference. In these practical scenarios, the ability to leverage pre-trained models on publicly available data (such as ZINC-15) to effectively tackle OOD problems represents a flexible and valuable approach. This aligns with our research objective of developing solutions that are applicable and beneficial in real-world settings.
>
> >Q3. It's not surprising that pretraining on a large molecular dataset can enhance a model's performance in OOD scenarios due to the semantic similarities within molecular datasets, similar to the way large language models operate. However, as seen in Table 2, pretraining on molecular graphs does not seem to generalize well to the CMNIST dataset. This highlights the limitations of pretraining methods
>
> The results in our paper support the idea that while pre-training may not generalize well in synthetic examples like the CMNIST dataset, it is highly effective in most real-world tasks, particularly in the molecular domain. The surprise lies in the fact that generically pre-trained models can outperform those specifically designed for OOD scenarios in these real-world applications. And importantly, our objective is to find how well a pre-trained model, trained on publicly available data, can handle downstream tasks in real-world scenarios, such as encountering molecules of different sizes or with different scaffold structures. If a pre-trained model demonstrates strong generalization in real-world molecular tasks, it provides a highly valuable approach to handling OOD problems in this domain. This is especially crucial in scenarios where labeled data are scarce and where the pre-trained model can leverage its learned representations to effectively address diverse and unseen challenges.
>
> ***References***
>
> [You et al., 2020] You, Yuning, Tianlong Chen, Yongduo Sui, Ting Chen, Zhangyang Wang, and Yang Shen. "Graph contrastive learning with augmentations." NeurIPS 2020.
>
> [Hou et al., 2022] Hou, Zhenyu, Xiao Liu, Yukuo Cen, Yuxiao Dong, Hongxia Yang, Chunjie Wang, and Jie Tang. "Graphmae: Self-supervised masked graph autoencoders." KDD 2022.

---

> ### Author Response · Authors · 2023-11-18
>
> >Q4. The analysis of factors influencing the performance of Pretraining methods in Section 4.3 appears somewhat shallow. For instance, the learning rate does not seem to be a factor worthy of analysis because it is not a critical component of the model. On the other hand, the author claims that pretrained models have a sample-efficient advantage. I initially thought this meant pretrained models can maintain good performance with smaller training sets. However, Figure 2b only demonstrates that pretrained models outperform baseline methods at different label rates.
>
> We would like to clarify these points as follows:
>
> - Significance of Learning Rate in Fine-Tuning: We acknowledge your concern regarding the emphasis on learning rate. However, the learning rate plays a significant role in fine-tuning the models. This is not only a practical consideration during fine-tuning but also a subject of theoretical research (e.g., Li et al., 2019).
>
> - Clarifying Sample Efficiency: our claim is based on the observation that pre-trained models maintain high performance on OOD tasks with only 10%-20% of the fine-tuning sample size typically required (not the label rate).
>
> **Questions**
> >This paper focuses merely on graph-level tasks. I wonder if pretrained methods can yield similar results on node-level OOD tasks.
>
> Yes, the current version primarily focuses on performance evaluation for chemical pre-trained models. We plan to introduce pretraining on social networks/citation network graphs to perform node-level comparisons in future work.
>
> ***Reference***
>
> [Li et al., 2019] Li, Yuanzhi, Colin Wei, and Tengyu Ma. "Towards explaining the regularization effect of initial large learning rate in training neural networks." NeurIPS  2019.

---

### Official Review · Reviewer_tP6E · 2023-10-31

**Soundness:** 3 good
**Presentation:** 2 fair
**Contribution:** 3 good
**Rating:** 5
**Confidence:** 4

**Summary:**

This manuscript explores the efficacy of pre-trained models in graph neural networks for out-of-distribution (OOD) scenarios. Through extensive experiments on various datasets, the authors discover that even basic pre-trained models perform as well as or outperform models specifically tailored to handle distribution shifts. The study also delves into the impact of critical factors like sample size and learning rates on the performance of pre-trained models. This research implies that pre-training can serve as a flexible and simple approach for OOD generalization in graph learning.

**Strengths:**

1. The paper conducts extensive experiments across diverse datasets, covering general and molecular graph domains, and varying degrees of distribution shift. This comprehensive approach provides a robust evaluation of the efficacy of pre-trained models in out-of-distribution (OOD) scenarios.
2. The findings of the paper show that even basic pre-trained models perform comparably or better than models specifically designed to handle distribution shift. This highlights the effectiveness of pre-training in graph OOD generalization and suggests that pre-training could be a flexible and simple approach for OOD generalization in graph learning.
3. The paper explores the influence of key factors such as sample size, learning rates, and in-distribution performance on the performance of pre-trained models for OOD generalization. This analysis provides insights into the factors that contribute to the effectiveness of pre-trained models in graph learning.

**Weaknesses:**

1. In the article, the discussion is confined to InfoGraph's appropriateness for general graph datasets that lack node information, and it does not include a comparative analysis of ContextPred, Attribute masking, and Mole-BERT. For example, in the MoleculeNet and OGBG-HIV datasets, there is asignificant performance gap between MOLE-BERT and CONTEXT-PRED.This absence of discussion regarding this phenomenon makes it challenging to determine which type of pre-trained model performs optimally in various scenarios.
2. This paper lacks an in-depth theoretical analysis of their examination of pre-trained models. This opens the door for further exploration of the fundamental principles and mechanisms behind the observed performance of pre-trained models in graph OOD scenarios.
3. The article lacks a detailed description of how OOD experiments with pre-trained models are conducted. It would be beneficial to use a figure to illustrate the framework.
4. In the introduction,  Please remove the extra ‘we’ in “Motivated by this potential, we we seek to investigate whether graph pre-trained models…".
5. In the introduction, this sentence “We observe that even with a smaller fine-tune sample size, such as only 10%-20%.....” can easily lead to ambiguity. It implies that pre-trained models using 10%-20% of the data can almost achieve the performance of the baselines that use all the data, rather than the performance achieved by pre-trained models using all the data.

**Questions:**

1. What are the reasons for different pre-trained models, such as ContextPred, Attribute masking, and Mole-BERT, yielding distinct results? What scenarios are each of them best suited for?
2. What is the framework for using pre-trained models in graph OOD tasks? Please describe the framework.
3. The sentence "We observe that even with a smaller fine-tune sample size, such as only 10%-20%..." in the introduction, does it have any ambiguity?

---

> ### Author Response · Authors · 2023-11-18
>
> We would like to thank the reviewer for the constructive comments and careful reading. The identified typo has been rectified, and the paper has been revised accordingly. The responses to the queries are outlined below.
>
> **Weakness**
> >Q1. In the article, the discussion is confined to InfoGraph's appropriateness for general graph datasets that lack node information, and it does not include a comparative analysis of ContextPred, Attribute masking, and Mole-BERT. For example, in the MoleculeNet and OGBG-HIV datasets, there is asignificant performance gap between MOLE-BERT and CONTEXT-PRED.This absence of discussion regarding this phenomenon makes it challenging to determine which type of pre-trained model performs optimally in various scenarios.
>
> We acknowledge the need for a detailed comparative analysis and explanation of the performance variations among different pre-trained models. Specifically, ContextPred focuses on predicting the surrounding graph structures of nodes within similar contexts, while Mole-BERT utilizes a context-aware tokenizer for encoding atoms, which can  be more effective in capturing the nuanced chemical properties essential for molecular datasets​​​​. In the meantime, the mask atom modeling design can also avoid the negative transfer in Attrimask. We have added this analysis in the revised version (See “General results” in section 4.3).
>
> >Q2. This paper lacks an in-depth theoretical analysis of their examination of pre-trained models. This opens the door for further exploration of the fundamental principles and mechanisms behind the observed performance of pre-trained models in graph OOD scenarios.
>
> Our study primarily concentrates on empirical investigations to understand the performance of various pre-training methods in graph OOD scenarios. It is very difficult to establish in-depth theory for large pre-trained models. The empirical results and observations from our study will provide empirical insights for future theoretical studies.
>
> >Q3. The article lacks a detailed description of how OOD experiments with pre-trained models are conducted. It would be beneficial to use a figure to illustrate the framework.
>
> Thank you for pointing this out. We have added Fig. A1.
>
> >Q5. In the introduction, this sentence “We observe that even with a smaller fine-tune sample size, such as only 10%-20%.....” can easily lead to ambiguity. It implies that pre-trained models using 10%-20% of the data can almost achieve the performance of the baselines that use all the data, rather than the performance achieved by pre-trained models using all the data.
>
> The pre-trained model achieves superior performance when fine-tuned with only 10%-20% of the sample size compared to the baseline method using the full sample size. We have updated this description in the revised paper to better clarify it.
>
> **Questions**
> >Q1. What are the reasons for different pre-trained models, such as ContextPred, Attribute masking, and Mole-BERT, yielding distinct results? What scenarios are each of them best suited for?
>
> Mole-BERT consistently outperforms or ranks second across the majority of molecular datasets. This is because, even though all three approaches leverage neighbor information from the center node to comprehensively capture chemical semantics,  Mole-BERT goes a step further than other methods introducing a context-aware tokenizer for encoding atoms, enhancing its ability to capture nuanced chemical properties crucial for molecular datasets. So for a practical recommendation, Mole-BERT is the most suitable pre-training methodology for molecular datasets.
>
> >Q2. What is the framework for using pre-trained models in graph OOD tasks? Please describe the framework.
>
> Please refer to response on Weakness 3.
>
> >Q3. The sentence "We observe that even with a smaller fine-tune sample size, such as only 10%-20%..." in the introduction, does it have any ambiguity?
>
> Please refer to response on Weakness 5.

---

### Official Review · Reviewer_RvDk · 2023-11-01

**Soundness:** 2 fair
**Presentation:** 3 good
**Contribution:** 2 fair
**Rating:** 3
**Confidence:** 4

**Summary:**

This paper studies how pre-training would impact GNNs' out-of-distribution(OOD) generalization performance. This work benchmarks four pre-training methods on multiple datasets with concept and covariate shifts. The experiments show that GNNs after pre-training would be more robust to OOD issues, and empirically they may perform comparable or better than those methods that are designed specifically to handle those shifts.

**Strengths:**

1. This paper benchmarks not only the OOD performance of multiple pre-training strategies but also some methods designed for handling distribution shifts, such as those rooted in invariant learning. It's interesting to put them together and compare them directly.
2. Multiple types of shifts are considered in the benchmark.

**Weaknesses:**

1. It seems to me the key observations are actually known, e.g., pre-training helps graph OOD generalization, especially for molecular tasks, and pre-training improves sample efficiency for labeled data. For example, [1] indeed has claimed their improvement on MoleculeNet with OOD splits, [2] has also summarized graph self-supervision techniques that help/claim OOD generalization. So, I think it's expected to see improvements.
2. I'm not sure how meaningful/resonable to directly compare pre-trained models with those models rooted in principles such as invariant learning. Their exact usage does not seem to be aligned and the level of information used is different.
3. If it is to serve as a comprehensive benchmark for pre-training methods for OOD generalization, I would expect more and newer methods with multiple backbone GNNs.

[1] Hu, Weihua, et al. "Strategies for pre-training graph neural networks." arXiv preprint arXiv:1905.12265 (2019).

[2] Li, Haoyang, et al. "Out-of-distribution generalization on graphs: A survey." arXiv preprint arXiv:2202.07987 (2022).

**Questions:**

- How did OGBG-HIV and OGBG-PCBA get split exactly for different shifts?

---

> ### Author Response · Authors · 2023-11-18
>
> Thank you for the insightful comments. Your feedback will certainly contribute to the improvement of our paper. Below, we address your concerns and questions.
>
> **Weakness**
> >Q1. It seems to me the key observations are actually known, e.g., pre-training helps graph OOD generalization, especially for molecular tasks, and pre-training improves sample efficiency for labeled data. For example, [1] indeed has claimed their improvement on MoleculeNet with OOD splits, [2] has also summarized graph self-supervision techniques that help/claim OOD generalization. So, I think it's expected to see improvements.
> [1] Hu, Weihua, et al. "Strategies for pre-training graph neural networks." arXiv preprint arXiv:1905.12265 (2019).
> [2] Li, Haoyang, et al. "Out-of-distribution generalization on graphs: A survey." arXiv preprint arXiv:2202.07987 (2022).
>
> Actually, we have already included these two works and highlighted key findings in Sec. 1. These two works indeed provided some preliminary analysis on graph pre-trained models for OOD generalization. However, they only include the OOD generalization results and analysis on MoleculeNet datasets with covariate shifts. In our work, we give a comprehensive analysis of graph pre-trained models for OOD generalization, ranging from a variety of data sources (DrugOOD, MoleculeNet, and OGBG, etc) and distribution shifts with varying degrees of shifts, together with different meta-analyses such as the effect of fine-tuning sample size and learning rates. We have updated our paper to better clarify this point (see “Graph Pre-training (2) Self-supervised Pre-training” in Sec. 2).
>
> >Q2. I'm not sure how meaningful/resonable to directly compare pre-trained models with those models rooted in principles such as invariant learning. Their exact usage does not seem to be aligned and the level of information used is different.
>
> It is true that the mechanisms and the level of information of pre-trained and invariant learning methods are somehow different. However, our goal is to explore the concept analogous to “foundation models”. The objective is not to discredit the importance of invariant learning but to assess the potential of pre-trained models in offering flexibility and practical value in OOD tasks. We found that generic pre-trained models without OOD as an objective can outperform specially designed approaches such as invariant learning. This suggests that pre-training could be a viable and efficient approach in graph learning for OOD challenges. The comparison aims to broaden the understanding of OOD generalization methods and highlight the practical applicability of pre-trained models in real-world scenarios.
>
> >Q3. If it is to serve as a comprehensive benchmark for pre-training methods for OOD generalization, I would expect more and newer methods with multiple backbone GNNs.
>
> We have incorporated two additional pre-trained methods, GraphCL(You et al., 2020) and GraphMAE (Hou et al., 2022), into our evaluations on the DrugOOD, MoleculeNet, and OGBG datasets, as presented in Table 2. Furthermore, we have introduced GAT (Velickovic et al., 2018) as a backbone on these datasets, with the corresponding results provided in Appendix D.3.
>
> **Questions**
> >How did OGBG-HIV and OGBG-PCBA get split exactly for different shifts?
>
> Regarding the two datasets split, we followed the methodology outlined in the GOOD benchmark (Gui et al., 2020).
>
> Specifically, for covariate shift with a distribution source of size, we arranged the molecules in descending order based on the number of nodes and split them into a ratio of 8:1:1 for the training set, validation set, and testing set, respectively. Similarly, the entire dataset was ordered based on the Bemis-Murcko scaffold string of SMILES, maintaining the same ratio.
>
> For concept shift, exemplified by size, we categorized molecules into different groups based on different numbers of molecular nodes. Following this categorization, we selected samples from each group with different labels, forming the training set, validation set, and testing set, respectively, with a ratio of 3:1:1. This grouping approach aligns with the scaffold-wise distribution, where molecules are categorized based on the Bemis-Murcko scaffold string of SMILES.
>
> ***References***
>
> [You et al., 2020] You, Yuning, Tianlong Chen, Yongduo Sui, Ting Chen, Zhangyang Wang, and Yang Shen. "Graph contrastive learning with augmentations." NeurIPS 2020.
>
> [Hou et al., 2022] Hou, Zhenyu, Xiao Liu, Yukuo Cen, Yuxiao Dong, Hongxia Yang, Chunjie Wang, and Jie Tang. "Graphmae: Self-supervised masked graph autoencoders." KDD 2022.
>
> [Velickovic et al., 2018] Petar Velickovic, Guillem Cucurull, Arantxa Casanova, Adriana Romero, Pietro Lio, and Yoshua
> Bengio. Graph attention networks. ICLR 2018.
>
> [Gui et al., 2022] Gui, Shurui, Xiner Li, Limei Wang, and Shuiwang Ji. "Good: A graph out-of-distribution benchmark." NeurIPS 2022.

---

> > ### Comment · Reviewer_RvDk · 2023-11-23
> >
> > Thanks for your responses. I find the key motivation of this benchmark remains less attractive to me, i.e., the weaknesses mentioned. I do appreciate the efforts put into benchmarking different methods across multiple datasets and settings, and it seems to me the key contribution of this work might be providing a unified evaluation of OOD performance across different methods. I have also read other reviews, and I intend to keep my initial score.

---

### Author Response · Authors · 2023-11-22
**We look forward to your further feedback**

Dear reviewers,

Thanks again for your insightful feedback on our work. As the discussion deadline approaches, we would like to know if our response has adequately addressed your concerns. In the rebuttal, we have highlighted the main motivations and significance to further clarify any doubts regarding our proposed benchmark. Moreover, we conducted more experiments, including evaluating two pre-trained models (GraphCL and GraphMAE) and employing GAT as another backbone model. We would like to know if our comments have fully addressed your concerns, and if not, we are eager to receive further feedback from you.

Best,

Authors

---

### Meta-Review · Area_Chair_xrsp · 2023-12-09

**Metareview:**

This work examines the effectiveness of pre-trained models in graph neural networks for out-of-distribution (OOD) tasks. Extensive experimentation on a variety of datasets reveals that basic pre-trained models can perform equally well, if not better, than models specially designed for handling OOD shifts. There is a compressive set of experiments and it is clear the authors spent a lot of time and effort on their paper. I think the paper has an interesting message that needs to be sharpened.

I believe the authors did not fully address reviewers' concerns regarding the assertion that pre-training sufficiently handles most OOD shifts; it is important to recognize that "OOD" is an umbrella term for what constitutes an infinite family of possible distribution shifts. For example, a closer look at Table 1 reveals that the datasets predominantly feature small average graph sizes, typical of molecular structures. Consequently, there can only be a relatively minor shift in graph sizes, particularly when considering variations in node degrees between training and testing.

I believe the work would have fared a lot better if the authors had contextualized their claims, and had made a clear statement of why their work is much more than "a more comprehensive analysis" of what had been already performed in previous works (as reviewer RvDk points out). For instance, if I have a test dataset, how can I know whether my pre-trained model is supposed to work well OOD for this specific dataset? Say, when is the size shift too much for a pre-trained model to handle?

Overall, I believe this is an interesting contribution. It needs a more clear differentiator with prior work and better contextualization of the conclusions, which are too broad for the experiments performed.

**Justification For Why Not Higher Score:**

The paper needs to be more than just a more comprehensive evaluation of previous work.

**Justification For Why Not Lower Score:**

N/A

---

### Decision · Program_Chairs · 2024-01-16

Reject